# On Graph Reconstruction via Empirical Risk Minimization: Fast Learning Rates and Scalability

**Guillaume Papa, Stéphan Clémençon**
LTCI, CNRS, Télécom ParisTech, Université Paris-Saclay
75013, Paris, France
`first.last@telecom-paristech.fr`

**Aurélien Bellet**
INRIA
59650 Villeneuve d'Ascq, France
`aurelien.bellet@inria.fr`

## Abstract

The problem of predicting connections between a set of data points finds many applications, in systems biology and social network analysis among others. This paper focuses on the *graph reconstruction* problem, where the prediction rule is obtained by minimizing the average error over all $n(n-1)/2$ possible pairs of the $n$ nodes of a training graph. Our first contribution is to derive learning rates of order $O_{\mathbb{P}}(\log n/n)$ for this problem, significantly improving upon the slow rates of order $O_{\mathbb{P}}(1/\sqrt{n})$ established in the seminal work of Biau and Bleakley (2006). Strikingly, these fast rates are *universal*, in contrast to similar results known for other statistical learning problems (*e.g.*, classification, density level set estimation, ranking, clustering) which require strong assumptions on the distribution of the data. Motivated by applications to large graphs, our second contribution deals with the computational complexity of graph reconstruction. Specifically, we investigate to which extent the learning rates can be preserved when replacing the empirical reconstruction risk by a computationally cheaper Monte-Carlo version, obtained by sampling with replacement $B \ll n^2$ pairs of nodes. Finally, we illustrate our theoretical results by numerical experiments on synthetic and real graphs.

## 1 Introduction

Although statistical learning theory mainly focuses on establishing *universal* rate bounds (*i.e.*, which hold for any distribution of the data) for the accuracy of a decision rule based on training observations, refined concentration inequalities have recently helped understanding conditions on the data distribution under which learning paradigms such as *Empirical Risk Minimization* (ERM) lead to faster rates. In binary classification, *i.e.*, the problem of learning to predict a random binary label $Y \in \{-1, +1\}$ from on an input random variable $X$ based on independent copies $(X_1, Y_1)$, ..., $(X_n, Y_n)$ of the pair $(X, Y)$, rates faster than $1/\sqrt{n}$ are achieved when little mass in the vicinity of $1/2$ is assigned by the distribution of the random variable $\eta(X) = \mathbb{P}\{Y = +1 \mid X\}$. This condition and its generalizations are referred to as the *Mammen-Tsybakov noise conditions* (see Mammen and Tsybakov, 1999; Tsybakov, 2004; Massart and Nédélec, 2006). It has been shown that a similar phenomenon occurs for various other statistical learning problems. Indeed, specific conditions under which fast rate results hold have been exhibited for density level set estimation (Rigollet and Vert, 2009), (bipartite) ranking (Clémençon et al., 2008; Clémençon and Robbiano, 2011; Agarwal, 2014), clustering (Antos et al., 2005; Clémençon, 2014) and composite hypothesis testing (Clémençon and Vayatis, 2010).

In this paper, we consider the supervised learning problem on graphs referred to as *graph reconstruction*, rigorously formulated by Biau and Bleakley (2006). The objective of graph reconstruction is to predict the possible occurrence of connections between a set of objects/individuals known to form the nodes of an undirected graph. Precisely, each node is described by a random vector $X$ which defines

a form of *conditional preferential attachment*: one predicts whether two nodes are connected based on their features $X$ and $X'$. This statistical learning problem is motivated by a variety of applications such as systems biology (*e.g.*, inferring protein-protein interactions or metabolic networks, see Jansen et al., 2003; Kanehisa, 2001) and social network analysis (*e.g.*, predicting future connections between users, see Liben-Nowell and Kleinberg, 2003). It has recently been the subject of a good deal of attention in the machine learning literature (see Vert and Yamanishi, 2004; Biau and Bleakley, 2006; Shaw et al., 2011), and is also known as *supervised link prediction* (Lichtenwalter et al., 2010; Cukierski et al., 2011). The learning task is formulated as the minimization of a *reconstruction risk*, whose natural empirical version is the average prediction error over the $n(n-1)/2$ pairs of nodes in a training graph of size $n$. Under standard complexity assumptions on the set of candidate prediction rules, excess risk bounds of the order $O_{\mathbb{P}}(1/\sqrt{n})$ for the empirical risk minimizers have been established by Biau and Bleakley (2006) based on a representation of the objective functional very similar to the *first Hoeffding decomposition* for second-order $U$-statistics (see Hoeffding, 1948). However, Biau & Bleakley ignored the computational complexity of finding an empirical risk minimizer, which scales at least as $O(n^2)$ since the empirical graph reconstruction risk involves summing up over $n(n-1)/2$ terms. This makes the approach impractical when dealing with large graphs commonly found in many applications.

Building up on the above work, our contributions to statistical graph reconstruction are two-fold:

**Universal fast rates.** We prove that a fast rate of order $O_{\mathbb{P}}(\log n/n)$ is always achieved by empirical reconstruction risk minimizers, in absence of any restrictive condition imposed on the data distribution. This is much faster than the $O_{\mathbb{P}}(1/\sqrt{n})$ rate established by Biau and Bleakley (2006). Our analysis is based on a different decomposition of the excess of reconstruction risk of any decision rule candidate, involving the *second Hoeffding representation* of a $U$-statistic approximating it, as well as appropriate maximal/concentration inequalities.

**Scaling-up ERM.** We investigate the performance of minimizers of computationally cheaper Monte-Carlo estimates of the empirical reconstruction risk, built by averaging over $B \ll n^2$ pairs of vertices drawn with replacement. The rate bounds we obtain highlight that $B$ plays the role of a tuning parameter to achieve an effective trade-off between statistical accuracy and computational cost. Numerical results based on simulated graphs and real-world networks are presented in order to support these theoretical findings.

The paper is organized as follows. In Section 2, we present the probabilistic setting for graph reconstruction and recall state-of-the-art results. Section 3 provides our fast rate bound analysis, while Section 4 deals with the problem of scaling-up reconstruction risk minimization to large graphs. Numerical experiments are displayed in Section 5, and a few concluding remarks are collected in Section 6. The technical proofs can be found in the Supplementary Material, along with some additional remarks and results.

## 2 Background and Preliminaries

We start by describing at length the probabilistic framework we consider for statistical inference on graphs, as introduced by Biau and Bleakley (2006). We then briefly recall the related theoretical results documented in the literature.

### 2.1 A Probabilistic Setup for Preferential Attachment

In this paper, $G = (V, E)$ is an undirected random graph with a set $V = \{1, \ldots, n\}$ of $n \geq 2$ vertices and a set $E = \{e_{i,j} : 1 \leq i \neq j \leq n\} \in \{0,1\}^{n(n-1)}$ describing its edges: for all $i \neq j$, we have $e_{i,j} = e_{j,i} = +1$ if the vertices $i$ and $j$ are connected by an edge and $e_{i,j} = e_{j,i} = 0$ otherwise. We assume that $G$ is a *Bernoulli graph*, *i.e.* the random variables $e_{i,j}, 1 \leq i < j \leq n$, are independent labels drawn from a Bernoulli distribution $Ber(p)$ with parameter $p = \mathbb{P}\{e_{i,j} = +1\}$, the probability that two vertices of $G$ are connected by an edge. The degree of each vertex is thus distributed as a binomial with parameters $n$ and $p$, which can be classically approximated by a Poisson distribution of parameter $\lambda > 0$ in the limit of large $n$, when $np \to \lambda$.

Whereas the marginal distribution of the graph $G$ is that of a Bernoulli graph (also sometimes abusively referred to as a *random graph*), a form of *conditional preferential attachment* is also specified in the framework considered here. Precisely, we assume that, for all $i \in V$, a continuous r.v.

$X_i$, taking its values in a separable Banach space $\mathcal{X}$, describes some features related to vertex $i$. The $X_i$'s are i.i.d. with common distribution $\mu(dx)$ and, for any $i \neq j$, the random pair $(X_i, X_j)$ models some information useful for predicting the occurrence of an edge connecting the vertices $i$ and $j$. Conditioned upon the features $(X_1, \ldots, X_n)$, any binary variables $e_{i,j}$ and $e_{k,l}$ are independent only if $\{i, j\} \cap \{k, l\} = \emptyset$. The conditional distribution of $e_{i,j}$, $i \neq j$, is supposed to depend on $(X_i, X_j)$ solely, described by the *posterior preferential attachment probability*:

$$\eta(X_i, X_j) = \mathbb{P}\{e_{i,j} = +1 \mid (X_i, X_j)\}. \tag{1}$$

For instance, $\forall(x_1, x_2) \in \mathcal{X}^2$, $\eta(x_1, x_2)$ can be a certain function of a specific distance or similarity measure between $x_1$ and $x_2$, as in the synthetic graphs described in Section 5.

The conditional average degree of the vertex $i \in V$ given $X_i$ (respectively, given $(X_1, \ldots, X_n)$) is thus $(n-1) \int_{x \in \mathcal{X}} \eta(X_i, x)\mu(dx)$ (respectively, $\sum_{j \neq i} \eta(X_i, X_j)$). Observe incidentally that, equipped with these notations, $p = \int_{(x,x') \in \mathcal{X}^2} \eta(x, x')\mu(dx)\mu(dx')$. Hence, the 3-tuples $(X_i, X_j, e_{i,j})$, $1 \leq i < j \leq n$, are *non-i.i.d.* copies of a generic random vector $(\mathbf{X}_1, \mathbf{X}_2, \mathbf{e}_{1,2})$ whose distribution $\mathcal{L}$ is given by the tensorial product $\mu(dx_1) \otimes \mu(dx_2) \otimes Ber(\eta(x_1, x_2))$, which is fully described by the pair $(\mu, \eta)$. Observe also that the function $\eta$ is symmetric by construction: $\forall(x_1, x_2) \in \mathcal{X}^2, \eta(x_1, x_2) = \eta(x_2, x_1)$.

In this framework, the learning problem introduced by Biau and Bleakley (2006), referred to as *graph reconstruction*, consists in building a symmetric *reconstruction rule* $g : \mathcal{X}^2 \to \{0, 1\}$, from a training graph $G$, with nearly minimum *reconstruction risk*

$$\mathcal{R}(g) = \mathbb{P}\{g(\mathbf{X}_1, \mathbf{X}_2) \neq \mathbf{e}_{1,2}\}, \tag{2}$$

thus achieving a comparable performance to that of the Bayes rule $g^*(x_1, x_2) = \mathbb{I}\{\eta(x_1, x_2) > 1/2\}$, whose risk is given by $\mathcal{R}^* = \mathbb{E}[\min\{\eta(\mathbf{X}_1, \mathbf{X}_2), 1 - \eta(\mathbf{X}_1, \mathbf{X}_2)\}] = \inf_g \mathcal{R}(g)$.

**Remark 1** (EXTENDED FRAMEWORK) *The results established in this paper can be straightforwardly extended to a more general framework, where $\mathcal{L} = \mathcal{L}^{(n)}$ may depend on the number $n$ of vertices. This allows to consider a general class of models, accounting for possible accelerating properties exhibited by various non scale-free real networks (Mattick and Gagen, 2005). An asymptotic study can be then carried out with the additional assumption that, as $n \to +\infty$, $\mathcal{L}^{(n)}$ converges in distribution to a probability measure $\mathcal{L}^{(\infty)}$ on $\mathcal{X} \times \mathcal{X} \times \{0, 1\}$, see (Biau and Bleakley, 2006). For simplicity, we restrict our study to the stationary case, i.e. $\mathcal{L}^{(n)} = \mathcal{L}$ for all $n \geq 2$.*

## 2.2 Related Results on Empirical Risk Minimization

A paradigmatic approach in statistical learning, referred to as *Empirical Risk Minimization* (ERM), consists in replacing (2) by its empirical version based on the labeled sample $\mathbb{D}_n = \{(X_i, X_j, e_{i,j}) : 1 \leq i < j \leq n\}$ related to $G$:[1]

$$\widehat{\mathcal{R}}_n(g) = \frac{2}{n(n-1)} \sum_{1 \leq i < j \leq n} \mathbb{I}\{g(X_i, X_j) \neq e_{i,j}\}. \tag{3}$$

An empirical risk minimizer $\widehat{g}_n$ is a solution of the optimization problem $\min_{g \in \mathcal{G}} \widehat{\mathcal{R}}_n(g)$, where $\mathcal{G}$ is a class of reconstruction rules of controlled complexity, hopefully rich enough to yield a small bias $\inf_{g \in \mathcal{G}} \mathcal{R}(g) - \mathcal{R}^*$. The performance of $\widehat{g}_n$ is measured by its excess risk $\mathcal{R}(\widehat{g}_n) - \inf_{g \in \mathcal{G}} \mathcal{R}(g)$, which can be bounded if we can derive probability inequalities for the maximal deviation

$$\sup_{g \in \mathcal{G}} |\widehat{\mathcal{R}}_n(g) - \mathcal{R}(g)|. \tag{4}$$

In the framework of classification, the flagship problem of statistical learning theory, the empirical risk is of the form of an average of i.i.d. r.v.'s, so that results pertaining to empirical process theory can be readily used to obtain bounds for the performance of empirical error minimization. Unfortunately, the empirical risk (3) is a sum of *dependent* variables. Following in the footsteps of Clémençon et al.

(2008), the work of Biau and Bleakley (2006) circumvents this difficulty by means of a representation of $\widehat{\mathcal{R}}_n(g)$ as an average of sums of i.i.d. r.v.'s, namely

$$\frac{1}{n!} \sum_{\sigma \in \mathfrak{S}_n} \frac{1}{\lfloor n/2 \rfloor} \sum_{i=1}^{\lfloor \frac{n}{2} \rfloor} \mathbb{I}\{g(X_{\sigma(i)}, X_{\sigma(i+\lfloor \frac{n}{2} \rfloor)}) \neq e_{\sigma(i),\sigma(i+\lfloor \frac{n}{2} \rfloor)}\},$$

where the sum is taken over all permutations of $\mathfrak{S}_n$, the symmetric group of order $n$, and $\lfloor u \rfloor$ denotes the integer part of any $u \in \mathbb{R}$. Very similar to the first Hoeffding decomposition for $U$-statistics (see Lee, 1990), this representation reduces the *first order* analysis of the concentration properties of (4) to the study of a basic empirical process (see Biau and Bleakley, 2006, Lemma 3.1). Biau and Bleakley (2006) thereby establish rate bounds of the order $O_{\mathbb{P}}(1/\sqrt{n})$ for the excess of reconstruction risk of $\widehat{g}_n$ under appropriate complexity assumptions (namely, $\mathcal{G}$ is of finite VC-dimension). Note incidentally that (3) is a $U$-statistic only when the variable $\eta(\mathbf{X}_1, \mathbf{X}_2)$ is almost-surely constant (see Janson and Nowicki, 1991, for an asymptotic study of graph reconstruction in this restrictive context).

**Remark 2** (ALTERNATIVE LOSS FUNCTIONS) *For simplicity, all our results are stated for the case of the 0-1 loss $\mathbb{I}\{g(X_i, X_j) \neq e_{i,j}\}$, but they straightforwardly extend to more practical alternatives such as the convex surrogate and cost-sensitive variants used in our numerical experiments. See the Supplementary Material for more details.*

## 3   Empirical Reconstruction is Always Fast!

In this section, we show that the rate bounds established by Biau and Bleakley (2006) can be largely improved *without* any additional assumptions. Precisely, we prove that fast learning rates of order $O_{\mathbb{P}}(\log n/n)$ are always attained by the minimizers of the empirical reconstruction risk (3), as revealed by the following theorem.

**Theorem 1** (FAST RATES) *Let $\widehat{g}_n$ be any minimizer of the empirical reconstruction risk (3) over a class $\mathcal{G}$ of finite VC-dimension $V < +\infty$. For all $\delta \in (0,1)$, we have w.p. at least $1 - \delta$: $\forall n \geq 2$,*

$$\mathcal{R}(\widehat{g}_n) - \mathcal{R}^* \leq 2 \left( \inf_{g \in \mathcal{G}} \mathcal{R}(g) - \mathcal{R}^* \right) + C \times \frac{V \log(n/\delta)}{n},$$

*where $C < +\infty$ is a universal constant.*[2]

**Remark 3** (ON THE BIAS TERM) *Apart from its remarkable universality, Theorem 1 takes the same form as in the case of empirical minimization of $U$-statistics (Clémençon et al., 2008, Corollary 6), with the same constant 2 in front of the bias term $\inf_{g \in \mathcal{G}} \mathcal{R}(g) - \mathcal{R}^*$. As can be seen from the proof, this constant has no special meaning and can be replaced by any constant strictly larger than 1 at the cost of increasing the constant $C$. Note that the $O_{\mathbb{P}}(1/\sqrt{n})$ rate obtained by Biau and Bleakley (2006) has a factor 1 in front of the bias term. Therefore, Theorem 1 provides a significant improvement unless the bias overly dominates the second term of the bound (i.e., the complexity of $\mathcal{G}$ is too small).*

**Remark 4** (ON COMPLEXITY ASSUMPTIONS) *We point out that a similar result can be established under weaker complexity assumptions involving Rademacher averages (refer to the Supplementary Material for more details). As may be seen by carefully examining the proof of Theorem 1, this would require to use the moment inequality for degenerate $U$-processes stated in (Clémençon et al., 2008, Theorem 11) instead of that proved by Arcones and Giné (1994).*

In the rest of this section, we outline the main ideas used to obtain this result (the detailed proofs can be found in the Supplementary Material). We rely on some arguments used in the fast rate analysis for empirical minimization of $U$-statistics (Clémençon et al., 2008), although these results only hold true under restrictive distributional assumptions. Whereas the quantity (3) is not a $U$-statistic, one may decompose the difference between the excess of reconstruction risk of any candidate rule $g \in \mathcal{G}$ and its empirical counterpart as the sum of its conditional expectation given the $X_i$'s, which is a $U$-statistic, plus a residual term. In order to explain the main argument underlying the present analysis, additional notation is required. Set

$$H_g(x_1, x_2, e_{1,2}) = \mathbb{I}\{g(x_1, x_2) \neq e_{1,2}\} \text{ and } q_g(x_1, x_2, e_{1,2}) = H_g(x_1, x_2, e_{1,2}) - H_{g^*}(x_1, x_2, e_{1,2})$$

for any $(x_1, x_2, e_{1,2}) \in \mathcal{X} \times \mathcal{X} \times \{0, 1\}$. Denoting by $\Lambda(g) = \mathcal{R}(g) - \mathcal{R}^* = \mathbb{E}[q_g(\mathbf{X}_1, \mathbf{X}_2, \mathbf{e}_{1,2})]$ the excess reconstruction risk with respect to the Bayes rule, its empirical estimate is given by

$$\Lambda_n(g) = \widehat{\mathcal{R}}_n(g) - \widehat{\mathcal{R}}_n(g^*) = \frac{2}{n(n-1)} \sum_{1 \leq i < j \leq n} q_g(X_i, X_j, e_{i,j}).$$

For all $g \in \mathcal{G}$, one may write:

$$\Lambda_n(g) - \Lambda(g) = U_n(g) + \widehat{W}_n(g), \qquad (5)$$

where

$$U_n(g) = \mathbb{E}\left[\Lambda_n(g) - \Lambda(g) \mid X_1, \ldots, X_n\right] = \frac{2}{n(n-1)} \sum_{1 \leq i < j \leq n} \widetilde{q}_g(X_i, X_j) - \Lambda(g)$$

is a $U$-statistic of degree 2 with symmetric kernel $\widetilde{q}_g(\mathbf{X}_1, \mathbf{X}_2) - \Lambda(g)$, where we denote $\widetilde{q}_g(\mathbf{X}_1, \mathbf{X}_2) = \mathbb{E}[q_g(\mathbf{X}_1, \mathbf{X}_2, \mathbf{e}_{1,2}) \mid \mathbf{X}_1, \mathbf{X}_2]$, and $\widehat{W}_n(g) = \frac{2}{n(n-1)} \sum_{i<j} \{q_g(X_i, X_j, e_{i,j}) - \widetilde{q}_g(X_i, X_j)\}$.

Equipped with this notation, we can now sketch the main steps of the proof of the fast rate bound stated in Theorem 1. As shown in the Supplementary Material, it is based on Eq. (5) combined with two intermediary results, each providing a control of one of the terms involved in it. The second order analysis carried out by Clémençon et al. (2008) shows that the small variance property of $U$-statistics may yield fast learning rates for empirical risk minimizers when $U$-statistics are used to estimate the risk, under a certain "low-noise" condition (see Assumption 4 therein). One of our main findings is that this condition is always fulfilled for the specific $U$-statistic $U_n(g)$ involved in the decomposition (5) of the excess of reconstruction risk of any rule candidate $g$, as shown by the following lemma.

**Lemma 2** (VARIANCE CONTROL) *For any distribution $\mathcal{L}$ and any reconstruction rule $g$, we have*

$$\mathrm{Var}\left(\mathbb{E}\left[q_g(\mathbf{X}_1, \mathbf{X}_2, \mathbf{e}_{1,2}) \mid \mathbf{X}_1\right]\right) \leq \Lambda(g).$$

The fundamental reason for the universal character of this result lies in the fact that the empirical reconstruction risk is not an average over all pairs (*i.e.*, a U-statistic of order 2) but an average over *randomly* selected pairs (random selection being ruled by the function $\eta$). The resulting smoothness is the key ingredient allowing us to establish the desired property.

Empirical reconstruction risk minimization over a class $\mathcal{G}$ being equivalent to minimization of $\Lambda_n(g) - \Lambda(g)$, the result below, combined with (5), proves that it also boils down to minimizing $U_n(g)$ under appropriate conditions on $\mathcal{G}$, so that the fast rate analysis of Clémençon et al. (2008) can be extended to graph reconstruction.

**Lemma 3** (UNIFORM APPROXIMATION) *Under the same assumptions as in Theorem 1, for any $\delta \in (0, 1)$, we have with probability larger than $1 - \delta$: $\forall n \geq 2$,*

$$\sup_{g \in \mathcal{G}} \left|\widehat{W}_n(g)\right| \leq C \times \frac{V \log(n/\delta)}{n},$$

*where $C < +\infty$ is a universal constant.*

The proof relies on classical symmetrization and randomization tricks combined with the *decoupling method*, in order to cope with the dependence structure of the variables and apply maximal/concentration inequalities for sums of independent random variables (see De la Pena and Giné, 1999).

Based on the above results, Theorem 1 can then be derived by relying on the *second Hoeffding decomposition* (see Hoeffding, 1948). This allows us to write $U_n(g)$ as a leading term taking the form of a sum of i.i.d r.v.'s with variance $4Var(\mathbb{E}[q_g(\mathbf{X}_1, \mathbf{X}_2, \mathbf{e}_{1,2}) \mid \mathbf{X}_1])$, plus a degenerate $U$-statistic (*i.e.*, a $U$-statistic of symmetric kernel $h(\mathbf{x}_1, \mathbf{x}_2)$ such that $\mathbb{E}[h(\mathbf{x}_1, \mathbf{X}_2)] = 0$ for all $\mathbf{x}_1 \in \mathcal{X}$). The latter can be shown to be of order $O_{\mathbb{P}}(1/n)$ uniformly over the class $\mathcal{G}$ by means of concentration results for degenerate $U$-processes.

We conclude this section by observing that, instead of estimating the reconstruction risk by (3), one could split the training dataset into two halves and consider the unbiased estimate of (2) given by

$$\frac{1}{\lfloor n/2 \rfloor} \sum_{i=1}^{\lfloor n/2 \rfloor} \mathbb{I}\{g(X_i, X_{i+\lfloor n/2 \rfloor}) \neq e_{i,i+\lfloor n/2 \rfloor}\}. \qquad (6)$$

The analysis of the generalization ability of minimizers of this empirical risk functional is simpler, insofar as only independent r.v.'s are involved in the sum (6). However, this estimate does not share the reduced variance property of (3) and although one could show that rate bounds of the same order as those stated in Theorem 1 may be attained by means of results pertaining to ERM theory for binary classification (see *e.g.* Section 5 in Boucheron et al., 2005), this would require a very restrictive assumption on distribution $\mathcal{L}$, namely to suppose that the posterior preferential attachment probability $\eta$ stays bounded away from $1/2$ with probability one (*cf* Massart and Nédélec, 2006). This is illustrated in the Supplementary Material.

## 4 Scaling-up Empirical Risk Minimization

The results of the previous section, as well as those of Biau and Bleakley (2006), characterize the excess risk achieved by minimizers of the empirical reconstruction risk $\widehat{\mathcal{R}}_n(g)$ but do not consider the computational complexity of finding such minimizers. For large training graphs, the complexity of merely computing $\widehat{\mathcal{R}}_n(g)$ is prohibitive as the number of terms involved in the summation is $O(n^2)$. In this section, we introduce a sampling-based approach to build approximations of the reconstruction risk with much fewer terms than $O(n^2)$, so as to scale-up risk minimization to large graphs.

The strategy we propose, inspired from the notion of *incomplete $U$-statistic* (see Lee, 1990), is of disarming simplicity: instead of the empirical reconstruction risk (3), we will consider an incomplete approximation obtained by sampling *pairs of vertices* (and not vertices) with replacement. Formally, we define the *incomplete graph reconstruction risk* based on $B \geq 1$ pairs of vertices as

$$\widetilde{\mathcal{R}}_B(g) = \frac{1}{B} \sum_{(i,j) \in \mathcal{P}_B} \mathbb{I}\{g(X_i, X_j) \neq e_{i,j}\}, \tag{7}$$

where $\mathcal{P}_B$ is a set of cardinality $B$ built by sampling with replacement in the set $\Theta_n = \{(i,j) : 1 \leq i < j \leq n\}$ of all pairs of vertices of the training graph $G$. For any $b \in \{1, \ldots, B\}$ and all $(i,j) \in \Theta_n$, denote by $\epsilon_b(i,j)$ the variable indicating whether the pair $(i,j)$ has been picked at the $b$-th draw ($\epsilon_b(i,j) = +1$) or not ($\epsilon_b(i,j) = +0$). The (multinomial) random vectors $\epsilon_b = (\epsilon_b(i,j))_{(i,j) \in \Theta_n}$ are i.i.d. (notice that $\sum_{(i,j) \in \Theta_n} \epsilon_b(i,j) = +1$ for $1 \leq b \leq B$) and the incomplete risk can be then rewritten as

$$\widetilde{\mathcal{R}}_B(g) = \frac{1}{B} \sum_{b=1}^{B} \sum_{(i,j) \in \Theta_n} \epsilon_b(i,j) \cdot \mathbb{I}\{g(X_i, X_j) \neq e_{i,j}\}. \tag{8}$$

Observe that the statistic (7) is an unbiased estimate of the true risk (2) and that, given the $X_i$'s, its conditional expectation is equal to (3). Considering (7) with $B = o(n^2)$ as our empirical risk estimate significantly reduces the computational cost, at the price of a slightly increased variance:

$$\text{Var}\left(\widetilde{\mathcal{R}}_B(g)\right) = \text{Var}\left(\widehat{\mathcal{R}}_n(g)\right) + \frac{1}{B}\left(\text{Var}\left(\widehat{\mathcal{R}}_1(g)\right) - \text{Var}\left(\widehat{\mathcal{R}}_n(g)\right)\right),$$

for any reconstruction rule $g$. Note in particular that the above variance is in general much smaller than that of the complete reconstruction risk based on a subsample of $\lfloor \sqrt{B} \rfloor$ vertices drawn at random (thus involving $O(B)$ pairs as well). We refer to the Supplementary Material for more details.

We are thus interested in characterizing the performance of solutions $\widetilde{g}_B$ to the computationally simpler problem $\min_{g \in \mathcal{G}} \widetilde{\mathcal{R}}_B(g)$. The following theorem shows that, when the class $\mathcal{G}$ is of finite VC-dimension, the concentration properties of the *incomplete reconstruction risk process* $\{\widetilde{\mathcal{R}}_B(g)\}_{g \in \mathcal{G}}$ can be deduced from those of the complete version $\{\widehat{\mathcal{R}}_n(g)\}_{g \in \mathcal{G}}$.

**Theorem 4** (UNIFORM DEVIATIONS) *Suppose that the class $\mathcal{G}$ is of finite VC-dimension $V < +\infty$. For all $\delta > 0$, $n \geq 1$ and $B \geq 1$, we have with probability at least $1 - \delta$:*

$$\sup_{g \in \mathcal{G}} |\widetilde{\mathcal{R}}_B(g) - \widehat{\mathcal{R}}_n(g)| \leq \sqrt{\frac{\log 2 + V \log\left((1 + n(n-1)/2)/\delta\right)}{2B}}.$$

The finite VC-dimension hypothesis can be relaxed and a bound of the same order can be proved to hold true under weaker complexity assumptions involving Rademacher averages (see Remark 4).

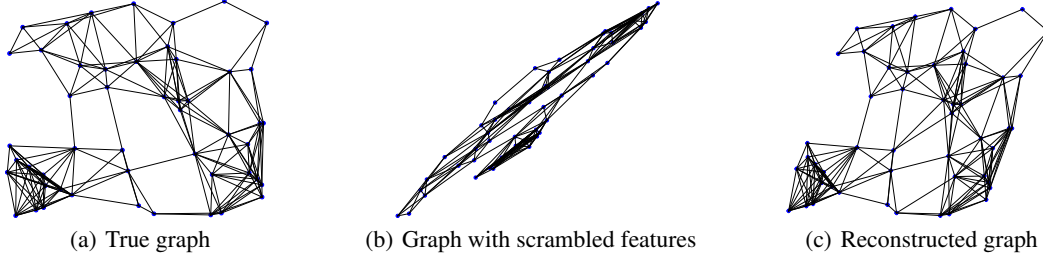

|(a) True graph | (b) Graph with scrambled features | (c) Reconstructed graph|

Figure 1: Illustrative experiment with $n = 50$, $q = 2$, $\tau = 0.27$ and $p = 0$. Figure 1(a) shows the training graph, where the position of each node is given by its 2D feature vector. Figure 1(b) depicts the same graph after applying a random transformation $R$ to the features. On this graph, the Euclidean distance with optimal threshold achieves a reconstruction error of 0.1311. In contrast, the reconstruction rule learned from $B = 100$ pairs of nodes (out of 1225 possible pairs) successfully inverts $R$ and accurately recovers the original graph (Figure 1(c)). Its reconstruction error is 0.008 on the training graph and 0.009 on a held-out graph generated with the same parameters.

Remarkably, with only $B = O(n)$ pairs, the rate in Theorem 4 is of the same order (up to a log factor) as that obtained by Biau and Bleakley (2006) for the maximal deviation $\sup_{g \in \mathcal{G}} |\widehat{\mathcal{R}}_n(g) - \mathcal{R}(g)|$ related to the complete reconstruction risk $\widehat{\mathcal{R}}_n(g)$ with $O(n^2)$ pairs. From Theorem 4, one can get a learning rate of order $O_{\mathbb{P}}(1/\sqrt{n})$ for the minimizer of the incomplete risk involving only $O(n)$ pairs.

Unfortunately, such an analysis does not exploit the relationship between conditional variance and expectation formulated in Lemma 2, and is thus not sufficient to show that reconstruction rules minimizing the incomplete risk (7) can achieve learning rates comparable to those stated in Theorem 1. In contrast, the next theorem provides sharper statistical guarantees. We refer to the Supplementary Material for the proof.

**Theorem 5** *Let $\widetilde{g}_B$ be any minimizer of the incomplete reconstruction risk* (7) *over a class $\mathcal{G}$ of finite VC-dimension $V < +\infty$. Then, for all $\delta \in (0,1)$, we have with probability at least $1 - \delta$: $\forall n \geq 2$,*

$$\mathcal{R}(\widetilde{g}_B) - \mathcal{R}^* \leq 2 \left( \inf_{g \in \mathcal{G}} \mathcal{R}(g) - \mathcal{R}^* \right) + CV \log(n/\delta) \times \left( \frac{1}{n} + \frac{1}{\sqrt{B}} \right),$$

*where $C < +\infty$ is a universal constant.*

This bound reveals that the number $B \geq 1$ of pairs of vertices plays the role of a tuning parameter, ruling a trade-off between statistical accuracy (taking $B(n) = O(n^2)$ fully preserves the convergence rate) and computational complexity. This will be confirmed numerically in Section 5.

The above results can be extended to other sampling techniques, such as Bernoulli sampling and sampling without replacement. We refer to the Supplementary Material for details.

## 5    Numerical Experiments

In this section, we present some numerical experiments on large-scale graph reconstruction to illustrate the practical relevance of the idea of incomplete risk introduced in Section 4. Following a well-established line of work (Vert and Yamanishi, 2004; Vert et al., 2007; Shaw et al., 2011), we formulate graph reconstruction as a distance metric learning problem (Bellet et al., 2015): we learn a distance function such that we predict an edge between two nodes if the distance between their features is smaller than some threshold. Assuming $\mathcal{X} \subseteq \mathbb{R}^q$, let $\mathbb{S}_+^q$ be the cone of symmetric PSD $q \times q$ real-valued matrices. The reconstruction rules we consider are parameterized by $M \in \mathbb{S}_+^q$ and have the form

$$g_M(x_1, x_2) = \mathbb{I}\{D_M(x_1, x_2) \leq 1\},$$

where $D_M(x_1, x_2) = (x_1 - x_2)^T M (x_1 - x_2)$ is a (pseudo) distance equivalent to the Euclidean distance after a linear transformation $L \in \mathbb{R}^{q \times q}$, with $M = L^T L$. Note that $g_M(x_1, x_2)$ can be seen as a linear separator operating on the pairwise representation $\text{vec}((x_1 - x_2)(x_1 - x_2')^T) \in \mathbb{R}^{q^2}$,

Table 1: Results (averaged over 10 runs) on synthetic graph with $n = 1,000,000$, $q = 100$, $p = 0.05$.

|  | **B = 0.01n** | **B = 0.1n** | **B = n** | **B = 5n** | **B = 10n** |
|---|---|---|---|---|---|
| **Reconstruction error** | 0.2272 | 0.1543 | 0.1276 | 0.1185 | 0.1159 |
| **Relative improvement** | – | 32% | 17% | 7% | 2% |
| **Training time (seconds)** | 21 | 398 | 5,705 | 20,815 | 42,574 |

hence the class of learning rules we consider has VC-dimension bounded by $q^2 + 1$. We define the reconstruction risk as:

$$\widehat{S}_n(g_M) = \frac{2}{n(n-1)} \sum_{i<j} [(2e_{i,j} - 1)(D_M(X_i, X_j) - 1)]_+ ,$$

where $[\cdot]_+ = \max(0, \cdot)$ is a convex surrogate for the 0-1 loss. In earlier work, ERM has only been applied to graphs with at most a few hundred or thousand nodes due to scalability issues. Thanks to our results, we are able to scale it up to much larger networks by sampling pairs of nodes and solve the resulting simpler optimization problem.

We create a synthetic graph with $n$ nodes as follows. Each node $i$ has features $X_i^{true} \in \mathbb{R}^q$ sampled uniformly over $[0, 1]$. We then add an edge between nodes that are at Euclidean distance smaller than some threshold $\tau$, and introduce some noise by flipping the value of $e_{i,j}$ for each pair of nodes $(i, j)$ independently with probability $p$. We then apply a random linear transformation $R \in \mathbb{R}^{q \times q}$ to each node to generate a "scrambled" version $X_i = RX_i^{true}$ of the nodes' features. The learning algorithm is only allowed to observe the scrambled features and must find a rule which accurately recovers the graph by solving the ERM problem above. Note that, denoting $D_{ij} = \|R^{-1}X_i - R^{-1}X_j\|_2$, the posterior preferential attachment probability is given by

$$\eta(X_i, X_j) = (1 - p) \cdot \mathbb{I}\{D_{ij} \leq \tau\} + p \cdot \mathbb{I}\{D_{ij} > \tau\}.$$

The process is illustrated in Figure 1. Using this procedure, we generate a training graph with $n = 1,000,000$ and $q = 100$. We set the threshold $\tau$ such that there is an edge between about 20% of the node pairs, and set $p = 0.05$. We also generate a test graph using the same parameters. We then sample uniformly with replacement $B$ pairs of nodes from the training graph to construct our incomplete reconstruction risk. The reconstruction error of the resulting empirical risk minimizer is estimated on 1,000,000 pairs of nodes drawn from the test graph. Table 1 shows the test error (averaged over 10 runs) as well as the training time for several values of $B$. Consistently with our theoretical findings, $B$ implements a trade-off between statistical accuracy and computational cost. For this dataset, sampling $B = 5,000,000$ pairs (out of $10^{12}$ possible pairs!) is sufficient to find an accurate reconstruction rule. A larger $B$ would result in increased training time for negligible gains in reconstruction error.

**Additional results.** In the Supplementary Material, we present comparisons to a node sampling scheme and to the "dataset splitting" strategy given by (6), as well as experiments on a real network.

## 6 Conclusion

In this paper, we proved that the learning rates for ERM in the graph reconstruction problem are always of order $O_{\mathbb{P}}(\log n/n)$. We also showed how sampling schemes applied to the population of edges (not nodes) can be used to scale-up such ERM-based predictive methods to very large graphs by means of a detailed rate bound analysis, further supported by empirical results. A first possible extension of this work would naturally consist in considering more general sampling designs, such as Poisson sampling (which generalizes Bernoulli sampling) used in graph sparsification (cf Spielman, 2005), and investigating the properties of minimizers of Horvitz-Thompson versions of the reconstruction risk (see Horvitz and Thompson, 1951). Another challenging line of future research is to extend the results of this paper to more complex unconditional graph structures in order to account for properties shared by some real-world graphs (*e.g.*, graphs with a power law degree distribution).

**Acknowledgments** This work was partially supported by the chair "Machine Learning for Big Data" of Télécom ParisTech and by a grant from CPER Nord-Pas de Calais/FEDER DATA Advanced data science and technologies 2015-2020.

## Footnotes

[1]A classical Lehmann-Scheffé argument shows that (3) is the estimator of (2) with smallest variance among all unbiased estimators.

[2]Note that, throughout the paper, the constant $C$ is not necessarily the same at each appearance.

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
