[Supplementary Material]

# On Graph Reconstruction via Empirical Risk Minimization: Fast Learning Rates and Scalability Supplementary Material

**Guillaume Papa, Stéphan Clémençon**
LTCI, CNRS, Télécom ParisTech, Université Paris-Saclay
75013, Paris, France
`first.last@telecom-paristech.fr`

**Aurélien Bellet**
INRIA
59650 Villeneuve d'Ascq, France
`aurelien.bellet@inria.fr`

This Supplementary Material is organized as follows. Section 1 contains some additional details on several points that were only briefly covered in the main text. Section 2 provides the detailed proofs of our results. Section 3 addresses alternative sampling schemes, and Section 4 presents additional experiments.

## 1 Additional Remarks

### 1.1 Alternative Loss Functions

For simplicity, our results are stated for the case of the classic 0-1 loss $\mathbb{I}\{g(X_i, X_j) \neq e_{i,j}\}$, but we point out that the same arguments directly apply to other loss functions (see Section 5.4 of Boucheron et al., 2005a, for instance). This includes the following two practical examples:

- In order to solve the ERM problem with efficient optimization methods (*e.g.* gradient-based), one typically considers a convex surrogate of the 0-1 loss (*e.g.*, hinge loss, logistic loss), as done in our experiments.
- Real-world networks can be very sparse (*i.e.*, they have very few edges), leading to a highly imbalanced prediction problem. One may then consider a *weighted* loss in order to assign a higher cost to errors on edges than to errors on non-edges.

### 1.2 On Fast Rates and the Noise Condition

As mentioned in the main text, instead of estimating the reconstruction risk by $\widehat{\mathcal{R}}_n(g)$, one could split the training dataset into two halves and consider the unbiased estimate of $\mathcal{R}(g)$ given by

$$\frac{1}{\lfloor n/2 \rfloor} \sum_{i=1}^{\lfloor n/2 \rfloor} \mathbb{I}\{g(X_i, X_{i+\lfloor n/2 \rfloor}) \neq e_{i,i+\lfloor n/2 \rfloor}\}. \tag{1}$$

Since only independent r.v.'s are involved in the sum (1), the analysis of its generalization ability is much simpler. In particular, fast rates can be obtained under the following classical noise condition (see Tsybakov, 2004; Boucheron et al., 2005b; Massart and Nédélec, 2006).

**Assumption 1.** *There exists $\beta > 0$ and $\alpha \in [0, 1]$ such that for all $t > 0$:*

$$\mathbb{P}\left( \left| \eta(\mathbf{X}_1, \mathbf{X}_2) - \frac{1}{2} \right| \leqslant t \right) \leqslant \beta t^{\alpha/(1-\alpha)}.$$

One can then show that minimizers of (1) achieve a learning rate of order $O_{\mathbb{P}}((\frac{\log n}{n})^{1/(2-\alpha)})$. We make the following observations:

- Assumption 1 is always satisfied for $\alpha = 0$ and corresponds to the classical learning rate of $O_{\mathbb{P}}(\sqrt{\log(n)/n})$ obtained by Biau and Bleakley (2006).

- Fast learning rates of the same order as the one we obtained for the minimizer of $\widehat{\mathcal{R}}_n(g)$ (see Theorem 1 of the main text) are achieved if and only if Assumption 1 is satisfied with $\alpha = 1$. This corresponds to the case where the posterior preferential attachment probability $\eta$ stays bounded away from $1/2$ with probability one (*cf* Massart and Nédélec, 2006).

In fact, the assumption $\alpha = 1$ is very restrictive. We illustrate this using the following toy example. Let $N \in \mathbb{N}^*$. For each node $1 \leqslant i \leqslant n$, we observe $X_i = (X_i^1, X_i^2)$, where $X_i^1$ and $X_i^2$ are two distinct elements drawn from $\{1, ..., N\}$. This may for instance correspond to the two preferred items of a user $i$ among a list of $N$ items. Consider now the case where two nodes are likely to be connected if they share common preferences, say $e_{i,j} \sim Ber(\#(X_i \cap X_j)/2)$. One can easily check that $\mathbb{P}(|\eta(\mathbf{X}_1, \mathbf{X}_2) - \frac{1}{2}| = 0) > 0$, so fast learning rates cannot be obtained for minimizers of (1). In contrast, the fast rates of Theorem 1 always hold for minimizers of $\widehat{\mathcal{R}}_n(g)$.

## 1.3 Sampling Nodes *vs.* Sampling Pairs of Nodes

In Section 4 of the main text, we promote an approach to scale-up graph reconstruction based on sampling $B$ pairs of nodes to build an *incomplete* reconstruction risk. Here, we give some insights on why this strategy should be preferred over sampling $m \geq 2$ nodes and building the *complete* reconstruction risk based on the resulting subgraph. To do so, we will compare their variance in the case where $B = m(m-1)/2$, so that both risk approximations consist of $B$ terms and thus have the same computational cost.

We first recall the definition of the incomplete reconstruction risk:

$$\widetilde{\mathcal{R}}_B(g) = \frac{1}{B} \sum_{(i,j) \in \mathcal{P}_B} \mathbb{I}\{g(X_i, X_j) \neq e_{i,j}\}, \tag{2}$$

where $\mathcal{P}_B$ is a set of cardinality $B$ built by sampling with replacement in the set $\Theta_n = \{(i,j) : 1 \leq i < j \leq n\}$ of all pairs of vertices of the training graph $G$. The variance of (2) is given by:

$$\mathrm{Var}\left(\widetilde{\mathcal{R}}_B(g)\right) = \mathrm{Var}\left(\widehat{\mathcal{R}}_n(g)\right) + \frac{1}{B}\left(\mathrm{Var}\left(\widehat{\mathcal{R}}_1(g)\right) - \mathrm{Var}\left(\widehat{\mathcal{R}}_n(g)\right)\right). \tag{3}$$

To characterize $\mathrm{Var}(\widetilde{\mathcal{R}}_B(g))$, we need to derive an explicit expression for $Var(\widehat{\mathcal{R}}_n(g))$. This is done by relying on the *second Hoeffding decomposition* (see Hoeffding, 1948) of $\widehat{\mathcal{R}}_n(g)$. For all $1 \leq i < j \leq n$, let us define

- $K_1(X_i) = \mathbb{E}[\mathbb{I}\{g(X_1, X_i) \neq e_{1,i}\}|X_i]$,
- $K_2(X_i, X_j) = \mathcal{R}(g) - K_1(X_i) - K_1(X_j) + \mathbb{E}[\mathbb{I}\{g(X_i, X_j) \neq e_{i,j}\}|X_i, X_j]$,
- $K_3(X_i, X_j, e_{i,j}) = \mathbb{I}\{g(X_i, X_j) \neq e_{i,j}\} - \mathbb{E}[\mathbb{I}\{g(X_i, X_j) \neq e_{i,j}\}|X_i, X_j]$.

We have the following "orthogonal" decomposition:

$$\widehat{\mathcal{R}}_n(g) - \mathcal{R}(g) = \frac{2}{n}\sum_{i=1}^n K_1(X_i) + \frac{2}{n(n-1)}\sum_{i<j} K_2(X_i, X_j) + \frac{2}{n(n-1)}\sum_{i<j} K_3(X_i, X_j, e_{i,j}).$$

Introducing the following variance terms:

- $\sigma_1^2 = \mathrm{Var}(K_1(X_1))$,
- $\sigma_2^2 = \mathrm{Var}(K_2(X_1, X_2))$,
- $\sigma_3^2 = \mathrm{Var}(K_3(X_1, X_2, e_{1,2}))$,

one easily gets

$$\mathrm{Var}(\widehat{\mathcal{R}}_n(g)) = \frac{4}{n}\sigma_1^2 + \frac{4}{n(n-1)}\left(\sigma_2^2 + \sigma_3^2\right).$$

Substituting the expression of $\mathrm{Var}(\widehat{\mathcal{R}}_n(g))$ in (3) gives

$$\mathrm{Var}\left(\widetilde{\mathcal{R}}_B(g)\right) = O\left(\max\left(\frac{1}{B}, \frac{1}{n}\right)\right).$$

This shows that if $B = O(n)$, $\mathrm{Var}(\widetilde{\mathcal{R}}_B(g))$ is of the same order as $\mathrm{Var}(\widehat{\mathcal{R}}_n(g))$ while $\widetilde{\mathcal{R}}_B(g)$ is computationally much cheaper than $\widehat{\mathcal{R}}_n(g)$ as it consists of only $O(n)$ terms.

In contrast to (2), the estimator obtained by sampling $m$ nodes has a larger variance: it is equal to $\mathrm{Var}(\widehat{\mathcal{R}}_m(g))$, which is of order $\frac{1}{m} = O(\frac{1}{\sqrt{B}})$.

In the additional numerical results presented in Section 4, we show that the performance gap between the two strategies is indeed significant in practice.

## 2 Technical Proofs

### 2.1 Proof of Lemma 2

For any reconstruction rule $g$, observe first that with probability one:

$$\begin{aligned}
\mathbb{E}[q_g(\mathbf{X}_1, \mathbf{X}_2, \mathbf{e}_{1,2}) \mid \mathbf{X}_1] &= \mathbb{E}[\mathbb{E}[q_g(\mathbf{X}_1, \mathbf{X}_2, \mathbf{e}_{1,2}) \mid \mathbf{X}_1, \mathbf{X}_2] \mid \mathbf{X}_1] \\
&= \mathbb{E}_{\mathbf{X}_2}[|1 - 2\eta(\mathbf{X}_1, \mathbf{X}_2)|\mathbb{I}\{g(\mathbf{X}_1, \mathbf{X}_2) \neq g^*(\mathbf{X}_1, \mathbf{X}_2)\}]]
\end{aligned}$$

Observing that we have

$$|1 - 2\eta(\mathbf{X}_1, \mathbf{X}_2)|^2 \leq |1 - 2\eta(\mathbf{X}_1, \mathbf{X}_2)|$$

almost surely, and combining with Jensen inequality, we have

$$\begin{aligned}
\mathrm{Var}(\mathbb{E}[q_g(\mathbf{X}_1, \mathbf{X}_2, \mathbf{e}_{1,2}) \mid \mathbf{X}_1]) &\leq \mathbb{E}_{\mathbf{X}_1}[(\mathbb{E}_{\mathbf{X}_2}[|1 - 2\eta(\mathbf{X}_1, \mathbf{X}_2)|\mathbb{I}\{g(\mathbf{X}_1, \mathbf{X}_2) \neq g^*(\mathbf{X}_1, \mathbf{X}_2)\}])^2] \\
&\leq \mathbb{E}[|1 - 2\eta(\mathbf{X}_1, \mathbf{X}_2)|\mathbb{I}\{g(\mathbf{X}_1, \mathbf{X}_2) \neq g^*(\mathbf{X}_1, \mathbf{X}_2)\}] \\
&= \Lambda(g).
\end{aligned}$$

### 2.2 Proof of Lemma 3

By definition, for all $g$, we have: $\forall n \geq 2$,

$$\widehat{W}_n(g) = \frac{2}{n(n-1)} \sum_{i<j} \{q_g(X_i, X_j, e_{i,j}) - \widetilde{q}_g(X_i, X_j)\}.$$

The proof relies on the key property: for all $i \neq j$,

$$\mathbb{E}[q_g(X_i, X_j, e_{i,j}) - \widetilde{q}_g(X_i, X_j)|X_i] = \mathbb{E}[q_g(X_i, X_j, e_{i,j}) - \widetilde{q}_g(X_i, X_j)|X_j] = 0$$

almost surely. This basically implies that the process $\{\widehat{W}_n(g)\}_{g \in \mathcal{G}}$ "behaves" as a second order Rademacher Chaos. Mimicking the techniques introduced in De la Pena and Giné (1999), this can be deduced from the following two technical lemmas, which we prove separately in Section 2.3 and Section 2.4 for clarity.

**Lemma 4.** (DECOUPLING) *Let* $(X_i')_{i=1}^n$ *be an independent copy of the sequence* $(X_i)_{i=1}^n$. *Consider r.v.'s valued in* $\{0, 1\}$, $\{\tilde{e}_{i,j}, i < j\}$, *conditionally independent given the* $X_i$'s *and the* $X_j'$'s *and such that* $\mathbb{P}(\tilde{e}_{i,j} = 1|X_i, X_j') = \eta(X_i, X_j')$. *Then, for all* $q \geq 1$, *we have:*

$$\mathbb{E}[\sup_{g \in \mathcal{G}} |\sum_{i<j} q_g(X_i, X_j, e_{i,j}) - \widetilde{q}_g(X_i, X_j)|^q] \leqslant 4^q \mathbb{E}[\sup_{g \in \mathcal{G}} |\sum_{i<j} q_g(X_i, X_j', \tilde{e}_{i,j}) - \widetilde{q}_g(X_i, X_j')|^q].$$

Thanks to the decoupling argument above, one can next introduce the following randomization.

**Lemma 5.** *Let* $(\sigma_i)_{i=1}^n$ *and* $(\sigma_i')_{i=1}^n$ *be two independent sequences of i.i.d. Rademacher variables, independent from the* $(X_i, X_i', e_{i,j}, \tilde{e}_{i,j})$'s. *Then, for all* $q \geq 1$, *we have:*

$$\mathbb{E}[\sup_{g \in \mathcal{G}} |\sum_{i<j} q_g(X_i, X_j', \tilde{e}_{i,j}) - \widetilde{q}_g(X_i, X_j')|^q] \leqslant 4^q \mathbb{E}[\sup_{g \in \mathcal{G}} |\sum_{i<j} \sigma_i \sigma_j'(q_g(X_i, X_j', \tilde{e}_{i,j}) - \widetilde{q}_g(X_i, X_j'))|^q].$$

Consider the conditional Rademacher average

$$\mathbb{E}_{\sigma,\sigma'}[\sup_{g\in\mathcal{G}}|\sum_{i<j}\sigma_i\sigma_j'(q_g(X_i,X_j',\tilde{e}_{i,j})-\tilde{q}_g(X_i,X_j'))|^q],$$

where $\mathbb{E}_{\sigma,\sigma'}$ denotes the expectation taken w.r.t. the $(\sigma_i,\sigma_i')$'s. Following Clémençon et al. (2008), we can derive an exponential inequality using Markov's inequality and show that, w.p. at least $1-\delta$,

$$\sup_{g\in\mathcal{G}}|\sum_{i<j}\sigma_i\sigma_j'\hat{h}_g(X_i,X_j',\tilde{e}_{i,j})| \leqslant C\times\frac{V\log(n/\delta)}{n}.$$

## 2.3 Proof of Lemma 4

For any random variable $\xi$, we denote by $\mathcal{L}(\xi)$ its distribution. Let $(X_i')_{i=1}^n$ be an independent copy of $(X_i)_{i=1}^n$, and $\mathcal{F}$ (respectively $\mathcal{F}'$) be the sigma-field generated by $\{X_1,...,X_n\}$ (respectively $\{X_1',...,X_n'\}$). Let $\{e_{i,j}', 1\leqslant i<j\leqslant n\}$ be Bernoulli random variables such that $\mathbb{P}(e_{i,j}'=1|\mathcal{F},\mathcal{F}')=\eta(X_i',X_j')$ (*i.e*, the conditional distribution of $e_{i,j}'$ depends on $(X_i',X_j')$ only). As in De la Pena and Giné (1999), let $(\sigma_i)_{i=1}^n$ be independent Rademacher variables and define:

$$Z_i = X_i \text{ if } \sigma_i = 1 \text{ and } X_i' \text{ otherwise,}$$
$$Z_i' = X_i' \text{ if } \sigma_i = 1 \text{ and } X_i \text{ otherwise.}$$

Conditionally upon the $X_i$ and $X_i'$, the random vector $(Z_i, Z_i')$ takes the values $(X_i, X_i')$ or $(X_i', X_i)$, each with probability 1/2. In particular, we have (see De la Pena and Giné, 1999):

$$\mathcal{L}(X_1,...,X_n,X_1',...,X_n') = \mathcal{L}(Z_1,...,Z_n,Z_1',...,Z_n'). \tag{4}$$

Let $\{\tilde{e}_{i,j}, 1\leqslant i<j\leqslant n\}$ be Bernoulli random variables such that $\mathbb{P}(\tilde{e}_{i,j}=1|\mathcal{F},\mathcal{F}')=\eta(X_i,X_j')$ and define for $i<j$:

$$\hat{e}_{i,j} = \begin{cases} e_{i,j} & \text{if } \sigma_i=1 \text{ and } \sigma_j=-1 \\ e_{i,j}' & \text{if } \sigma_i=-1 \text{ and } \sigma_j=1 \\ \tilde{e}_{i,j} & \text{if } \sigma_i=1 \text{ and } \sigma_j=1 \\ \tilde{e}_{j,i} & \text{if } \sigma_i=-1 \text{ and } \sigma_j=1. \end{cases}$$

We also recall the following notations:

$$\begin{aligned} H_g(x_1,x_2,e_{1,2}) &= \mathbb{I}\{g(x_1,x_2)\neq e_{1,2}\}, \\ q_g(x_1,x_2,e_{1,2}) &= H_g(x_1,x_2,e_{1,2})-H_{g^*}(x_1,x_2,e_{1,2}) \\ \tilde{q}_g(X_1,X_2) &= \mathbb{E}[q_g(X_1,X_2,e_{1,2})|X_1,X_2] \end{aligned}$$

Let $\hat{h}_g = q_g - \tilde{q}_g$ and notice that for all $i<j$:

$$\mathbb{E}_\sigma[\hat{h}_g(Z_i,Z_j',\hat{e}_{i,j})] = \frac{1}{4}\Big(\hat{h}_g(X_i,X_j,e_{i,j})+\hat{h}_g(X_i',X_j',e_{i,j}')+\hat{h}_g(X_i,X_j',\tilde{e}_{i,j})+\hat{h}_g(X_i',X_j,\tilde{e}_{j,i})\Big),$$

where $\mathbb{E}_\sigma$ denotes the expectation taken with respect to $\sigma_1,...,\sigma_n$. Moreover, using

$$\mathbb{E}[\hat{h}_g(X_i',X_j',e_{i,j}')|\mathcal{F}] = \mathbb{E}[\hat{h}_g(X_i',X_j',e_{i,j}')] = 0$$

and

$$\mathbb{E}[\hat{h}_g(X_i,X_j',\tilde{e}_{i,j})|\mathcal{F}] = \mathbb{E}[q_g(X_i,X_j',\tilde{e}_{i,j})|X_i] - \mathbb{E}[\mathbb{E}[q_g(X_i,X_j',\tilde{e}_{i,j})|X_i,X_j']|X_i] = 0,$$

we easily get

$$\hat{h}_g(X_i,X_j,e_{i,j}) = 4\mathbb{E}[\hat{h}_g(Z_i,Z_j',\hat{e}_{i,j})]|\mathcal{F}].$$

For all $q>1$, we therefore have:

$$\mathbb{E}[\sup_{g\in\mathcal{G}}|\sum_{i<j}\hat{h}_g(X_i,X_j,e_{i,j})|^q] \leqslant 4^q\mathbb{E}[\sup_{g\in\mathcal{G}}|\sum_{i<j}\hat{h}_g(Z_i,Z_j',\hat{e}_{i,j})|^q].$$

We now use (4) combined with the fact that by construction, the law of $\hat{e}_{i,j}$ only depends on the realizations $Z_i, Z_j'$, *i.e.*, $\mathbb{P}(\hat{e}_{i,j}=1|Z_i,Z_j')=\eta(Z_i,Z_j')$, to obtain

$$\mathbb{E}[\sup_{g\in\mathcal{G}}|\sum_{i<j}\hat{h}_g(Z_i,Z_j',\hat{e}_{i,j})|^q] = \mathbb{E}[\sup_{g\in\mathcal{G}}|\sum_{i<j}\hat{h}_g(X_i,X_j',\tilde{e}_{i,j})|^q],$$

which concludes the proof.

## 2.4  Proof of Lemma 5

In this section, we find it more convenient to work with sums over $\{1 \leqslant i \neq j \leqslant n\}$ than sums over $\{1 \leqslant i < j \leqslant n\}$, so that for $i < j$ and any random variables $a_{i,j}$, we set $a_{j,i} = a_{i,j}$. Using the symmetry of our problem we have:

$$2^q \mathbb{E}[\sup_{g \in \mathcal{G}} | \sum_{i<j} \hat{h}_g(X_i, X'_j, \tilde{e}_{i,j})|^q] = \mathbb{E}[\sup_{g \in \mathcal{G}} | \sum_{i \neq j} \hat{h}_g(X_i, X'_j, \tilde{e}_{i,j})|^q],$$

where $\hat{h}_g$ is defined as in Section 2.3. Re-using the notations used in Section 2.3, we further introduce $(X''_i)_{i=1}^n$, a copy of $(X'_i)_{i=1}^n$, independent from $\mathcal{F}$, $\mathcal{F}'$, and denote by $\mathcal{F}''$ its sigma-field. Let $\{\tilde{e}''_{i,j}, 1 \leqslant i < j \leqslant n\}$ Bernoulli random variables such that $\mathbb{P}(\tilde{e}''_{i,j} = 1 | \mathcal{F}, \mathcal{F}', \mathcal{F}'') = \eta(X_i, X''_j)$. We now use classical randomization techniques and introduce our "ghost" sample:

$$
\begin{aligned}
\mathbb{E}[\sup_{g \in \mathcal{G}} | \sum_{i \neq j} \hat{h}_g(X_i, X'_j, \tilde{e}_{i,j})|^q] &= \mathbb{E}[\sup_{g \in \mathcal{G}} | \sum_{i \neq j} \hat{h}_g(X_i, X'_j, \tilde{e}_{i,j}) - \mathbb{E}_{\mathcal{F}''}[\hat{h}_g(X_i, X''_j, \tilde{e}''_{i,j})]|^q] \\
&\leqslant \mathbb{E}\left\{ \sup_{g \in \mathcal{G}} \left| \sum_{j=1}^n \sum_{i \neq j} \hat{h}_g(X_i, X'_j, \tilde{e}_{i,j}) - \hat{h}_g(X_i, X''_j, \tilde{e}''_{i,j}) \right|^q \right\}.
\end{aligned}
$$

where $\mathbb{E}_{\mathcal{F}''}$ denotes expectation with respect to the $(X''_i)_{i=1}^n$. Let $(\sigma_i)_{i=1}^n$ be independent Rademacher variables, independent of $\mathcal{F}$, $\mathcal{F}'$ and $\mathcal{F}''$, then we have:

$$
\begin{aligned}
\mathbb{E}[\sup_{g \in \mathcal{G}} | \sum_{j=1}^n \sum_{i \neq j} \hat{h}_g(X_i, X'_j, \tilde{e}_{i,j}) &- \hat{h}_g(X_i, X''_j, \tilde{e}''_{i,j})|^q \mathcal{F}] \\
&= \mathbb{E}[\sup_{g \in \mathcal{G}} | \sum_{j=1}^n \sigma_j \sum_{i \neq j} \hat{h}_g(X_i, X'_j, \tilde{e}_{i,j}) - \hat{h}_g(X_i, X''_j, \tilde{e}''_{i,j})|^q | \mathcal{F}] \\
&\leqslant 2^q \mathbb{E}[\sup_{g \in \mathcal{G}} | \sum_{j=1}^n \sigma_j \sum_{i \neq j} \hat{h}_g(X_i, X'_j, \tilde{e}_{i,j})|^q | \mathcal{F}],
\end{aligned}
$$

and get:

$$\mathbb{E}[\sup_{g \in \mathcal{G}} | \sum_{i \neq j} \hat{h}_g(X_i, X'_j, \tilde{e}_{i,j})|^q] \leqslant 2^q \mathbb{E}[\sup_{g \in \mathcal{G}} | \sum_{j=1}^n \sigma_j \sum_{i \neq j} \hat{h}_g(X_i, X'_j, \tilde{e}_{i,j})|^q].$$

We now repeat the same argument but for the $(X_i)_{i=1}^n$. Let $(X'''_i)_{i=1}^n$ be a copy of $(X_i)_{i=1}^n$, independent of $\mathcal{F}$, $\mathcal{F}'$, and denote by $\mathcal{F}'''$ its sigma-field. Let $\{\tilde{e}'''_{i,j}, 1 \leqslant i < j \leqslant n\}$ be Bernoulli random variables such that $\mathbb{P}(\tilde{e}''_{i,j} = 1 | \mathcal{F}, \mathcal{F}', \mathcal{F}''') = \eta(X'''_i, X'_j)$. Then:

$$
\begin{aligned}
\mathbb{E}[\sup_{g \in \mathcal{G}} | \sum_{j=1}^n \sigma_j \sum_{i \neq j} \hat{h}_g(X_i, X'_j, \tilde{e}_{i,j})|^q] &= \mathbb{E}\Big[ \sup_{g \in \mathcal{G}} | \sum_{i=1}^n \sum_{j \neq i} \sigma_j \hat{h}_g(X_i, X'_j, \tilde{e}_{i,j}) \\
&\qquad\qquad - \mathbb{E}_{\mathcal{F}'''}[\sigma_j \hat{h}_g(X'''_i, X'_j, \tilde{e}''_{i,j})]|^q \Big] \\
&\leqslant \mathbb{E}\Big[ \sup_{g \in \mathcal{G}} | \sum_{i=1}^n \sum_{j \neq i} \sigma_j \hat{h}_g(X_i, X'_j, \tilde{e}_{i,j}) \\
&\qquad\qquad - \sigma_j \hat{h}_g(X'''_i, X'_j, \tilde{e}''_{i,j})|^q \Big].
\end{aligned}
$$

Let $(\sigma'_i)_{i=1}^n$ be a copy of $(\sigma_i)_{i=1}^n$, independent of $(\sigma_i)_{i=1}^n$, $\mathcal{F}$, $\mathcal{F}'$, $\mathcal{F}'''$, we have

$$
\begin{aligned}
\mathbb{E}[\sup_{g \in \mathcal{G}} | \sum_{j=1}^n \sum_{i \neq j} \sigma_j \hat{h}_g(X_i, X'_j, \tilde{e}_{i,j}) &- \sigma_j \hat{h}_g(X'''_i, X'_j, \tilde{e}''_{i,j})|^q | \mathcal{F}', (\sigma_i)_{i=1}^n] \\
&= \mathbb{E}[\sup_{g \in \mathcal{G}} | \sum_{i=1}^n \sigma'_i \sum_{j \neq i} \sigma_j \hat{h}_g(X_i, X'_j, \tilde{e}_{i,j}) - \sigma_j \hat{h}_g(X_i, X''_j, \tilde{e}''_{i,j})|^q | \mathcal{F}', (\sigma_i)_{i=1}^n] \\
&\leqslant 2^q \mathbb{E}[\sup_{g \in \mathcal{G}} | \sum_{i=1}^n \sigma'_i \sum_{j \neq i} \sigma_j \hat{h}_g(X_i, X'_j, \tilde{e}_{i,j})|^q | \mathcal{F}', (\sigma_i)_{i=1}^n].
\end{aligned}
$$

Finally, we get

$$\mathbb{E}[\sup_{g \in \mathcal{G}} | \sum_{j=1}^{n} \sum_{i \neq j} \hat{h}_g(X_i, X'_j, \tilde{e}_{i,j})|^q] \leqslant 4^q \mathbb{E}[\sup_{g \in \mathcal{G}} | \sum_{i \neq j} \sigma_i \sigma'_j \hat{h}_g(X_i, X'_j, \tilde{e}_{i,j})|^q].$$

## 2.5 Proof of Theorem 1

We prove a more general version of Theorem 1.

**Theorem 6.** *For any $\theta > 0$, with probability at least $1 - \delta$, the empirical risk minimizer $\hat{g}_n$ satisfies:*

$$\mathcal{R}(\hat{g}_n) - \mathcal{R}(g') \leqslant \theta(\mathcal{R}(g') - \mathcal{R}(g^*)) + (1 + \theta + \frac{1}{\theta})DV\frac{\log(n/\delta)}{n}$$

*for some universal constant D.*

The version of the main text is obtained by taking $\theta = 1$ and adding $\mathcal{R}(g') - \mathcal{R}(g^*)$ on both sides of the inequality.

*Proof.* Following the analysis of Clémençon et al. (2008), for all $g \in \mathcal{G}$ we rewrite:

$$\Lambda_n(g) - \Lambda(g) = 2(T_n(g) - \Lambda(g)) + W_n(g) + \widehat{W}_n(g).$$

where

$$T_n(g) = \frac{1}{n} \sum_{i=1}^{n} h_g(X_i)$$

is a sum of i.i.d random variables with $h_g(X_i) = \mathbb{E}[q_g(X_i, X_j, e_{i,j})|X_i]$,

$$W_n(g) = \frac{1}{n(n-1)} \sum_{i \neq j} \widetilde{h}_g(X_i, X_j)$$

is a degenerate U-statistic with symmetric kernel

$$\widetilde{h}_g(X_i, X_j) = \mathbb{E}[q_g(X_i, X_j, e_{i,j})|X_i, X_j] + \Lambda(g) - h_g(X_i) - h_g(X_j)$$

and

$$\widehat{W}_n(g) = \frac{1}{n(n-1)} \sum_{i \neq j} \hat{h}_g(X_i, X_j, e_{i,j})$$

with

$$\hat{h}_g(X_i, X_j, e_{i,j}) = q_g(X_i, X_j, e_{i,j}) - \mathbb{E}[q_g(X_i, X_j, e_{i,j})|X_i, X_j].$$

We also recall that we proved in Lemma 2 that

$$\text{Var}\left(\mathbb{E}\left[q_g(\mathbf{X}_1, \mathbf{X}_2, \mathbf{e}_{1,2}) \mid \mathbf{X}_1\right]\right) \leq \Lambda(g).$$

As mentioned before, the fact that we can upper-bound the variance of $h_g(X)$ by its expectation is the key property that will allow us to derive fast rates. We now follow the analysis of Boucheron et al. (2005a) and introduce the following quantities:

- $\mathcal{H} = \{h_g : g \in \mathcal{G}\}$.

- $\mathcal{H}^* = \{\alpha h_g : g \in \mathcal{G}, \alpha \in [0, 1]\}$.

- $\psi(r) = \mathbb{E}R_n\left\{f \in \mathcal{H}^* : \sqrt{\mathbb{E}[f(X)^2]} \leqslant r\right\}$, where $R_n$ denotes the Rademacher chaos taken over the observation $X_i$.

- For $r > 0$, we note $\mathcal{G}_r = \left\{\frac{rh_g}{\max(r, \sqrt{\Lambda(g)})} : g \in \mathcal{G}\right\}$.

For all $h \in \mathcal{G}_r$, $\mathbb{E}[h] - h \leqslant 2$ and $\mathrm{Var}(h) \leqslant r$ so that applying Bousquet's Inequality for the Supremum of Empirical Processes Bousquet (2002); Boucheron et al. (2005a) to the class $\mathcal{G}_r$ gives that with probability at least $1 - \delta/6$, for any $g \in \mathcal{G}$:

$$\Lambda(g) - T_n(g) \leqslant \frac{\max(r, \sqrt{\Lambda(g)})}{r} \Big( 2\mathbb{E}[\sup_{h_g \in \mathcal{G}_r} \Lambda(g) - T_n(g)] + r\sqrt{\frac{2\log(6/\delta)}{n}} + \frac{8\log(1/\delta)}{3n} \Big).$$

Since $\mathbb{E}[\sup_{h_g \in \mathcal{G}_r}(\Lambda(g) - T_n(g))] \leqslant 2\mathbb{E}R_n[G_r] \leqslant 2\psi(r)$ we get:

$$\Lambda(g) - T_n(g) \leqslant \frac{\max(r, \sqrt{\Lambda(g)})}{r} \Big( 4\psi(r) + r\sqrt{\frac{2\log(6/\delta)}{n}} + \frac{8\log(6/\delta)}{3n} \Big).$$

We now apply Bernstein's Inequality to $h'_g$ and using the fact that $\mathrm{Var}(h'_g) \leqslant \Lambda(g') \leqslant \Lambda(g)$ for any $g \in \mathcal{G}$, we get that with probability at least $1 - \delta/6$:

$$T_n(g') - \Lambda(g') \leqslant \max(r, \sqrt{\Lambda(g)})\sqrt{\frac{2\log(6/\delta)}{n}} + \frac{8\log(6/\delta)}{3n}.$$

Summing the two inequality and taking a union bound we get that with probability at least $1 - \delta/3$, for all $g \in \mathcal{G}$:

$$\Lambda(g) - T_n(g) + T_n(g') - \Lambda(g') \leqslant \frac{\max(r, \sqrt{\Lambda(g)})}{r} \Big( 4\psi(r) + 2r\sqrt{\frac{2\log(6/\delta)}{n}} + \frac{16\log(6/\delta)}{3n} \Big).$$

We now rewrite $T_n(g)$ as:

$$T_n(g) = \frac{1}{2}(\Lambda(g) + \Lambda_n(g) - W_n(g) - \widehat{W}_n(g))$$

which we substitute in the previous inequality and obtain:

$$\frac{3}{2}(\Lambda(g) - \Lambda(g')) \leqslant \frac{1}{2}(\Lambda_n(g) - \Lambda_n(g')) + W_n(g') - W_n(g) + \widehat{W}_n(g') - \widehat{W}_n(g)$$
$$+ \frac{\max(r, \sqrt{\Lambda(g)})}{r} \Big( 4\psi(r) + 2r\sqrt{\frac{2\log(6/\delta)}{n}} + \frac{16\log(6/\delta)}{3n} \Big).$$

We take $g = \widehat{g}_n$ so that $\Lambda_n(\widehat{g}_n) - \Lambda_n(g') \leqslant 0$ and use Lemma 3 together with a result from Clémençon et al. (2008) to obtain that with probability at least $1 - 2\delta/3$:

$$W_n(g') - W_n(g) + \widehat{W}_n(g') - \widehat{W}_n(g) \leqslant 2\sup_{g \in \mathcal{G}}|W_n(g)| + 2\sup_{g \in \mathcal{G}}|\widehat{W}_n(g)| \leqslant \frac{4CV\log(3n/\delta)}{n}.$$

We finally get that with probability at least $1 - \delta$:

$$\Lambda(\widehat{g}_n) - \Lambda(g') \leqslant \frac{4CV\log(3n/\delta)}{n} + \frac{\max(r, \sqrt{\Lambda(\widehat{g}_n)})}{r} \times \Big( 4\psi(r) + 2r\sqrt{\frac{2\log(6/\delta)}{n}} + \frac{16\log(6/\delta)}{3n} \Big).$$

Now, we either have $\Lambda(\widehat{g}_n) \leqslant r^2$, in which case we have in particular $\Lambda(\widehat{g}_n) - \Lambda(g') \leqslant r^2$, or $\Lambda(\widehat{g}_n) \geqslant r^2$. Under the latter hypothesis:

$$\Lambda(\widehat{g}_n) - \Lambda(g') \leqslant \frac{4CV\log(3n/\delta)}{n} + \frac{\sqrt{\Lambda(\widehat{g}_n)}}{r} \Big( 4\psi(r) + 2r\sqrt{\frac{2\log(6/\delta)}{n}} + \frac{16\log(6/\delta)}{3n} \Big).$$

For $\delta \in [0,1]$, we finally introduce $r^*(\delta)$ as solution of

$$r = 4\psi(\sqrt{r}) + 2\sqrt{r}\sqrt{\frac{2\log(6/\delta)}{n}} + \frac{16\log(6/\delta)}{3n}.$$

Substituting $r^*(\delta)^2$ for $r$ and using its definition in the previous bound gives:

$$\Lambda(\widehat{g}_n) - \Lambda(g') \leqslant \frac{4CV\log(3n/\delta)}{n} + \sqrt{\Lambda(\widehat{g}_n)r^*(\delta)}.$$

Now, using for all $\theta > 0$:

$$\sqrt{\Lambda(\widehat{g}_n)r^*(\delta)} \leqslant \frac{1}{2}\Big( \frac{2\theta}{1+\theta}\Lambda(\widehat{g}_n) + \frac{1+\theta}{2\theta}r^*(\delta) \Big)$$

gives that with probability at least $1 - \delta$:

$$\mathcal{R}(\widehat{g}_n) - \mathcal{R}(g') \leqslant \theta(\mathcal{R}(g') - \mathcal{R}(g^*)) + \frac{(\theta+1)^2}{4\theta}r^*(\delta) + (1+\theta)4CV\log(3n/\delta).$$

Putting all the pieces back together, we have shown

$$\mathcal{R}(\widehat{g}_n) - \mathcal{R}(g') \leqslant \max(r^*(\delta)^2, \theta(\mathcal{R}(g') - \mathcal{R}(g^*)) + \frac{(\theta+1)^2}{4\theta}r^*(\delta) + (1+\theta)4CV\log(3n/\delta)).$$

Convenient upperbound for $r^*(\delta)$ can be found in Boucheron et al. (2005a):

$$r^*(\delta) \leqslant CV\frac{log(n/\delta)}{n},$$

for some universal constant $C$. This concludes the proof. $\qquad\square$

## 2.6 Proof of Theorem 4

One may write for all $g \in \mathcal{G}$, $n \geq 2$ and $B \geq 1$,

$$\widetilde{\mathcal{R}}_B(g) - \widehat{\mathcal{R}}_n(g) = \frac{1}{B}\sum_{b=1}^{B}\mathcal{Z}_b(g),$$

where

$$\mathcal{Z}_b(g) = \sum_{i<j}\left(\epsilon_b(i,j) - \frac{2}{n(n-1)}\right)\mathbb{I}\{g(X_i, X_j) \neq e_{i,j}\}$$

for all $(g,b) \in \mathcal{G} \times \{1,\ \ldots,\ B\}$. Conditioned upon the $(X_i, X_j, e_{i,j})$'s, for all $g \in \mathcal{G}$, the $\mathcal{Z}_b(g)$'s are i.i.d. centered random variables, bounded by $1$. In addition, the collection $\mathcal{G}$ being of finite VC-dimension $V$, Sauer's lemma yields:

$$\#\{\{\mathbb{I}\{g(X_i, X_j) \neq e_{i,j}\}:\ g \in \mathcal{G}\} \leq (1 + n(n-1)/2)^V.$$

Applying Hoeffding's inequality to the $\mathcal{Z}_b(g)$'s conditioned upon the $(X_i, X_j, e_{i,j})$'s and the union bound leads to: $\forall \zeta > 0$,

$$\mathbb{P}\left\{\sup_{g \in \mathcal{G}}\left|\frac{1}{B}\sum_{b=1}^{B}\mathcal{Z}_b(g)\right| > \zeta \mid \{(X_i, X_j, e_{i,j})\}_{(i,j)\in\Lambda}\right\} \leq 2(1 + n(n-1)/2)^V \exp\left(-2B\zeta^2\right).$$

Taking the expectation w.r.t. the $(X_i, X_j, e_{i,j})$'s yields the desired bound.

## 2.7 Proof of Theorem 5

As done for Theorem 1, we prove the following generalization of Theorem 5.

**Theorem 5.** *For any $\theta > 0$, with probability at least $1 - \delta$, the minimizer $\widetilde{g}_B$ of the incomplete risk satisfies:*

$$\mathcal{R}(\widetilde{g}_B) - \mathcal{R}(g') \leqslant \theta(\mathcal{R}(g') - \mathcal{R}(g^*)) + (1 + \theta + \frac{1}{\theta})D_1V\left(\frac{\log(n/\delta)}{n} + \sqrt{\frac{\log(n/\delta)}{B}}\right).$$

*for some universal constant $D_1$.*

*Proof.* We proceed in a similar fashion than for the proof of Theorem 6 and first start by recalling that we have with probability at least $1 - \delta/4$,

$$\sup_{g \in \mathcal{G}}|W_n(g)| \leqslant \frac{CV\log(4n/\delta)}{n}$$

and

$$\sup_{g \in \mathcal{G}}|\widehat{W}_n(g)| \leqslant \frac{CV\log(4n/\delta)}{n}.$$

We also recall that Theorem 4 gives that with probability at least $1 - \delta/4$:

$$\sup_{g \in \mathcal{G}} |\widetilde{\Lambda}_B(g) - \widehat{\Lambda}_n(g)| \le \sqrt{\frac{\log 2 + V \log\left(\frac{4(1+n(n-1)/2)}{\delta}\right)}{2B}} := \sqrt{\frac{C_1 V \log(4n/\delta)}{B}}.$$

We follow the proof of Theorem 6. For all $r > 0$ with probability at least $1 - \delta/4$:

$$\Lambda(g) - T_n(g) + T_n(g') - \Lambda(g') \le \frac{\max(r, \sqrt{\Lambda(g)})}{r}\left(4\psi(r) + \frac{16\log(8/\delta)}{3n} + 2r\sqrt{\frac{2\log(8/\delta)}{n}}\right).$$

For any $g \in \mathcal{G}$, let $\widetilde{\Lambda}_B(g) = \widetilde{\mathcal{R}}_B(g) - \widetilde{\mathcal{R}}_B(\widetilde{g}_B)$ be the incomplete excess risk of $g$. We rewrite:

$$T_n(g') - T_n(g) = \frac{1}{2}(\Lambda(g') - \Lambda(g) + \Lambda_n(g') - \widetilde{\Lambda}_B(g') + \widetilde{\Lambda}_B(g') - \widetilde{\Lambda}_B(g) + \widetilde{\Lambda}_B(g) - \Lambda_n(g)$$
$$+ W_n(g') - W_n(g) + \widehat{W}_n(g') - \widehat{W}_n(g)), \quad (5)$$

which we substitute in the previous bound, take $g = \widetilde{g}_B$ so that $\widetilde{\Lambda}_B(\widetilde{g}_B) - \widetilde{\Lambda}_B(g') \le 0$ and get that with probability at least $1 - \delta$:

$$\Lambda(\widetilde{g}_B) - \Lambda(g') \le \frac{\max(r, \sqrt{\Lambda(g)})}{r}\left(4\psi(r) + 2r\sqrt{\frac{2\log(8/\delta)}{n}} + \frac{16\log(8/\delta)}{3n}\right)$$
$$+ 4CV \log(4n/\delta) + \sqrt{\frac{C_1 V \log(4n/\delta)}{B}}.$$

Let $r_1^*(\delta)$ be solution of

$$r = 4\psi(r) + 2r\sqrt{\frac{2\log(8/\delta)}{n}} + \frac{16\log(8/\delta)}{3n}.$$

Then we either have $\Lambda(\widetilde{g}_B) \le r_1^*(\delta)^2$ or:

$$\mathcal{R}(\widetilde{g}_B) - \mathcal{R}(g') = \Lambda(\widetilde{g}_B) - \Lambda(g') \le \sqrt{\Lambda(\widetilde{g}_B) r_1^*(\delta)} + 4CV \log(4n/\delta) + \sqrt{\frac{C_1 V \log(4n/\delta)}{B}}.$$

In the latter case we easily get that for all $\theta > 0$,

$$\mathcal{R}(\widetilde{g}_B) - \mathcal{R}(g') \le \theta(\mathcal{R}(g') - \mathcal{R}(g^*)) + \frac{(\theta+1)^2}{4\theta} r_1^*(\delta) + (1+\theta)\Big(4CV \log(4n/\delta)$$
$$+ \sqrt{\frac{C_1 V \log(4n/\delta)}{B}}\Big).$$

Upper-bounding $r_1^*(\delta)$ as in the proof of Theorem 6 gives the result. $\qquad\square$

## 3 Extensions to Alternative Sampling Schemes

In this section, we show that the results of the previous section can be extended to other sampling techniques, such as Bernoulli sampling and sampling without replacement. Borrowing the terminology of survey theory, the (finite) population under study is the collection $\Theta_n$ of all pairs of vertices of the graph $G$. Its cardinality is $\#\Theta_n = n(n-1)/2$. In this context, a *survey sample* is any subset $S$ of $\Theta_n$ with (possibly random) cardinality $m \le n(n-1)/2$, referred to as its *size*. A survey scheme without replacement is thus any conditional probability distribution $\mathcal{D}$ on the power set of $\Theta_n$, $\mathcal{P}(\Theta_n)$, given the data $\mathbb{D}_n = \{(X_i, X_j, e_{i,j}) : (i,j) \in \Theta_n\}$. The probability that the pair $(i,j) \in \Theta_n$ belongs to the sample $S$ drawn from $\mathcal{D}$, conditioned upon $\mathbb{D}_n$, is denoted by $\pi_{(i,j)} = \mathbb{P}_{\mathcal{D}}\{(i,j) \in S\}$ and termed a first order inclusion probability. The quantities $\pi_{(i,j),(k,l)} = \mathbb{P}_{\mathcal{D}}\{((i,j),(k,l)) \in S^2\}$, for $(i,j) \ne (k,l)$, are referred to as second order inclusion probabilities. Equipped with these notations, the Horvitz-Thompson version (Horvitz and Thompson, 1951) of the empirical reconstruction risk of a rule $g \in \mathcal{G}$ based on a sample $S \sim \mathcal{D}$ is then given by:

$$\widetilde{\mathcal{R}}^{(\mathcal{D})}(g) = \frac{2}{n(n-1)} \sum_{(i,j) \in \Lambda} \frac{\epsilon_{i,j}}{\pi_{i,j}} \mathbb{I}\{g(X_i, X_j) \ne e_{i,j}\}, \quad (6)$$

where $\epsilon_{i,j} = \mathbb{I}\{(i,j) \in S\}$ for all $(i,j) \in \Theta_n$ and $0/0 = 0$ by convention. Provided that the $\pi_{(i,j)}$'s are all strictly positive, (6) is an unbiased estimate of $\widehat{\mathcal{R}}_n(g)$ and, when the size $B \leq n(n-1)/2$ of the survey scheme is deterministic, its conditional variance given the training graph is $\mathrm{Var}(\widetilde{\mathcal{R}}^{(\mathcal{D})}(g) \mid \mathbb{D}_n) = 4/(n(n-1))^2 \times \sum_{(i,j) \neq (k,l)} \sigma^2_{(i,j),(k,l)}$, where $\sigma^2_{(i,j),(k,l)}$ is given by

$$\left( \frac{\mathbb{I}\{g(X_i, X_j) \neq e_{i,j}\}}{\pi_{(i,j)}} - \frac{\mathbb{I}\{g(X_k, X_l) \neq e_{k,l}\}}{\pi_{(k,l)}} \right)^2 \times \left( \pi_{(i,j),(k,l)} - \pi_{(i,j)}\pi_{(k,l)} \right)$$

for all $(i,j) \neq (k,l)$. Two specific sampling techniques can naturally be considered.

**Bernoulli sampling.** Let $B \leq n(n-1)/2$. This sampling plan corresponds to the situation where the $\epsilon_{(i,j)}$'s are i.i.d. Bernoulli r.v.'s with parameter $2B/(n(n-1))$. In this case, the (random) size is a binomial variable of size $n(n-1)/2$ with $B$ as expected value. Incidentally, we mention that Bernoulli sampling is a particular case of Poisson sampling (relaxing the assumption that the $\epsilon_{i,j}$'s are identically distributed), widely used for the purpose of graph sparsification (see *e.g.* Spielman, 2005, Section 6).

**Sampling without replacement (SWOR).** Fixing in advance $B \leq n(n-1)/2$, one may uniformly draw a sample $S$ among all samples of size $B$ (there are $\binom{n(n-1)/2}{B}$ such samples). In this case, we have $\pi_{(i,j)} = 2B/(n(n-1))$ and $\pi_{(i,j),(k,l)} = 4B(B-1)/(n(n-1)^2(n-2))$ for all $(i,j) \neq (k,l)$ in $\Theta_n$. This is a special case of *rejective sampling*, corresponding to the situations where the $\pi_{(i,j)}$'s are all equal.

The following proposition reveals that, just like (2), the Horvitz-Thompson reconstruction risk (6), when based on SWOR or Bernoulli schemes, estimates the empirical reconstruction risk of rules in $\mathcal{G}$ uniformly well (provided that $\mathcal{G}$ is of finite VC-dimension).

**Proposition 6.** *Suppose that $\mathcal{G}$ is of finite VC-dimension $V < +\infty$. For all $\delta \in (0,1)$, we have with probability larger than $1 - \delta$: for all $n \geq 1$, $B \leq n(n-1)/2$,*

$$\sup_{g \in \mathcal{G}} |\widetilde{\mathcal{R}}^{(\mathcal{D})}(g) - \widehat{\mathcal{R}}_n(g)| \leq \sqrt{\frac{2\log\left( \frac{2(1+n(n-1)/2)^V}{\delta} \right)}{B}},$$

*if $\mathcal{D}$ is a Bernoulli plan of expected size $B \leq n(n-1)/2$,*

$$\sup_{g \in \mathcal{G}} |\widetilde{\mathcal{R}}^{(\mathcal{D})}(g) - \widehat{\mathcal{R}}_n(g)| \leq \frac{2\log\left( \frac{2(1+n(n-1)/2)^V}{\delta} \right)}{B} + \sqrt{\frac{2\log\left( \frac{2(1+n(n-1)/2)^V}{\delta} \right)}{B}},$$

*when $\mathcal{D}$ is a SWOR plan of size $B \leq n(n-1)/2$.*

*Proof.* For the Bernoulli case, we apply Bernstein inequality to the sum $Z(g)$ of the r.v.'s

$$\left( \epsilon_{i,j} - \frac{2B}{n(n-1)} \right) \mathbb{I}\{g(X_i, X_j) \neq e_{i,j}\}$$

conditioned upon the graph $\mathbb{D}_n$, which are bounded by 1 and have conditional variance less than $2B/(n(n-1))$. We obtain: $\forall g \in \mathcal{G}, \forall \zeta > 0$,

$$\mathbb{P}\{|Z(g)| > \zeta \mid \mathbb{D}_n\} \leq 2\exp\left( -\frac{\zeta^2}{2B + 2\zeta/3} \right).$$

Using the union bound, one gets: $\forall \zeta > 0$,

$$\mathbb{P}\left\{ \sup_{g \in \mathcal{G}} |Z(g)| > B\zeta \mid \mathbb{D}_n \right\} \leq 2\exp\left( -\frac{B\zeta^2}{2 + 2\zeta/3} \right).$$

Solving $\delta = 2(1 + n(n-1)/2)^d \exp(-B\zeta^2/(2 + 2\zeta/3))$ leads to the first bound.

Turning next to the second bound, the exponential inequality tailored to the SWOR case (see Serfling, 1974, Corollary 1.1) yields:

$$\mathbb{P}\left\{ \frac{1}{B}|Z(g)| > \zeta \mid \mathbb{D}_n \right\} \leq 2\exp\left( -\frac{B\zeta^2}{2} \right),$$

for all $g \in \mathcal{G}, \zeta > 0$. Using the union bound and then solving $\delta = 2(1 + n(n-1)/2)^d \exp(-B\zeta^2/2)$ completes the proof. $\qquad\square$

Table 1: Reconstruction error on synthetic graph with parameters $n = 1,000,000$, $q = 100$, $p = 0.05$.

| | B = 0.01n | B = 0.1n | B = n | B = 5n | B = 10n |
|---|---|---|---|---|---|
| **Sampling nodes** | 0.2552 | 0.1847 | 0.1411 | 0.1279 | 0.1233 |
| **Sampling pairs of nodes** | 0.2272 | 0.1543 | 0.1276 | 0.1185 | 0.1159 |

Figure 1: Summary of results on the synthetic graph.

Note that following Section 2.7, one can easily derive a version of Theorem 5 for the minimizer of $\widetilde{\mathcal{R}}^{\mathcal{D}}$ by replacing $\widetilde{\Lambda}_B(g)$ with $\widetilde{\Lambda}_B^{\mathcal{D}}(g)$ (the incomplete excess risk corresponding to the sampling plan $\mathcal{D}$) in the decomposition (5) and making the appropriate modifications.

## 4 Additional Numerical Experiments

In this section, we first present some additional results on the synthetic graph reconstruction problem introduced in the main text. We then display some experiments on a real citation network.

### 4.1 Synthetic Graph

In Section 4 of the main text, we proposed and analyzed a strategy to scale up graph reconstruction, which consists in approximating the reconstruction risk by sampling $B$ pairs of nodes. This strategy was empirically shown to be effective in experiments on a synthetic graph reconstruction task (Section 5).

Here, we compare the above scheme with an alternative strategy based on sampling *nodes* mentioned in Section 1.3. Specifically, we randomly sample $m$ nodes and use the reconstruction risk evaluated on the resulting subgraph as an approximation to the risk on the full graph. To allow a fair comparison, we set $m$ such that $B \simeq m(m-1)/2$ in order to get approximately the same computational complexity for both approaches. Table 1 compares the reconstruction error of both strategies for various values of $B$, averaged over 10 runs. Sampling nodes leads to significantly larger error than sampling pairs, as it only leverages information from a subset of training nodes. We have noticed experimentally that the error gap between these two strategies widens as the complexity of the class of reconstruction rules gets larger (for instance, if we increase the data dimension $q$).

Figure 1 summarizes these results by displaying both the test error and the training time with respect to the number of terms in the risk estimate. For completeness, we also show the performance of the "dataset splitting" strategy (see Eq. 7 of the main text). It exhibits a good statistical/runtime trade-off but leads to suboptimal test error.

Table 2: Reconstruction error (averaged over 10 runs) on the Cit-HepTh graph.

|  | B = 10K | B = 100K | B = 1M | B = 5M |
|---|---|---|---|---|
| **Balanced reconstruction error** | 0.3080 | 0.2629 | 0.2484 | 0.2464 |
| **Relative improvement** | – | 15% | 6% | <1% |
| **Training time (seconds)** | 418 | 1,675 | 4,481 | 18,895 |

## 4.2 Real Network

Finally, we also validate our approach on Cit-HepTh, the high-energy physics theory citation network extracted from arXiv.[1] The graph has 27,770 nodes representing research papers and 352,807 edges corresponding to a citation between two papers. We generate simple features based on the paper abstracts as follows. We first remove stop words and those with less than 4 characters, then apply a tokenizer and stemmer from the NLTK library[2] and keep only the 300 most frequent words among all abstracts. Finally, we build a 300-dimensional bag-of-words feature vector for each paper by counting the number of occurrences of these words in its abstract and applying an $L_1$-norm normalization. We randomly split the nodes into a training set (80%) and a test set (20%). Note that the graph is very sparse: there is an edge between about 0.1% of the node pairs. Since classification error is not meaningful in such an imbalanced regime, we optimize a balanced error rate by sampling active edges with higher probability (this is equivalent to optimizing a weighted version of the reconstruction risk, see the remark at the beginning of Section 2). Note that the Euclidean distance (with threshold tuned on the training set) achieves a balanced test error of about 0.37.

Table 2 shows the balanced test error (averaged over 10 runs) as well as the training time for several values of $B$. Despite the higher dimensional and sparse nature of the features, we are able to significantly improve over the Euclidean baseline using few training pairs. Furthermore, sampling $B = 1\text{M}$ pairs is sufficient to get very close to the best performance: going from 1M to 5M pairs brings less than 1% relative improvement in test error at the expense of a 4 times increase in training time.

## Footnotes

[1]`http://snap.stanford.edu/data/cit-HepTh.html`

[2]`http://www.nltk.org`