[Reviews · NeurIPS 2016]

Reviewer 1

Summary

Establishes fast rates for link prediction in random networks where each node has an associated feature vector.

Qualitative Assessment

This is a very nice paper that should clearly be accepted unless there is some technical flaw that I have overlooked. Just a couple of minor comments: There are a few issues with the citations. Mammen and Tsybakov is 1999. Also Massart and Nedelec did not introduce the margin assumption. Perhaps you meant an earlier paper of Massart? I would also like to point the authors attention to the paper Ann. Statist. Volume 23, Number 3 (1995), 855-881. Measuring Mass Concentrations and Estimating Density Contour Clusters-An Excess Mass Approach Wolfgang Polonik In Thm 3.6 the margin assumption also appears. To my knowledge this is actually the first appearance of this type of assumption. Please update your literature review accordingly. In lines 127 – 128, how can a 2006 paper follow in the footsteps of a 2008 paper? Eqn (5), \widehat{W}_n is only defined implicitly, please define it explicitly. In the statement of theorem 4, you have for all n and for all B inside the with high probability statement. Yet the form of the bound does not suggest that you have applied a union bound. Please comment.

Confidence in this Review

2-Confident (read it all; understood it all reasonably well)


Reviewer 2

Summary

This paper considers a generative model of a random graph. Graphs are generated as follows. We start with a set of vertices. For each vertex, we independently and identically sample a feature from a vector space according to a law \mu. A stochastic kernel, \eta, determines the Bernoulli probability of an edge being present, conditioned on two (unordered) vertex features. For each pair of vertices, we use \eta to sample the Bernoulli variable thus determining whether there is an edge or not. We have no a priori knowledge of \mu or \eta, except for a fully observed (features+edges) training graph with n vertices. In a partially observed (features only) test graph, we would like to infer where there are edges. We focus on a simplified test scenario, where two features (X1,X2) are sampled independently according to \mu, and an edge E is determined according to \eta(.|X1,X2). The learning goal is as follows: give the n-vertex training graph, construct a binary function g_n that maps two features into a label (edge present or absent). The performance of g is determined by the probability of error: R(g_n)=\Prob\{ g_n(X1,X2) \neq E \}. If \eta is known, the Bayes (best) risk-achieving g^* can be obtained. To avoid overfitting, one would generally restrict the model of possible gs (say to VC dimension V), and then compare to the benchmark of the best-in-model, g'. The paper shows that g_n obtained by empirical risk minimization is only O(log(n)/n) off from g', in the following sense: there is a constant b, such that for all a > 1, w.p. > 1-d: R(g_n)-R(g^*) \leq a (R(g')-R(g^*)) + (a+1/(a-1)) b V log(n/d)/n This establishes an effective convergence rate of O(log(n)/n), improving on the previously best known rate of O(1/sqrt(n)). The authors also propose a sub-sampling based method to scale the approach and give some numerical experiments to support the theory.

Qualitative Assessment

- The paper is generally very well written. The result is interesting and solid, and potentially impactful. It is however a bit strange that the authors sell their results from the perspective of "fast rates despite no margin conditions" (presumably because of the proof techniques), while the crux of it is that "dependent samples are almost independent". This is because the number of (albeit not independent) samples is not n, but rather N=n(n-1)/2, and therefore we have a "slow" rate for N. - When proposing the edge-sampling scaling technique, the authors claim that "sampling vertices directly would greatly damage the learning rate". The evidence that they give toward this is a variance comparison, on Line 63 in the supplementary material. First, the "is of order 1/m" claim on that line is not obvious (and probably incorrect). Second, if we forget sub-sampling and use (the even simpler) truncation method by discarding all but the first m vertices and the resulting sub-tree, then Theorem 1 applies as is, and we get an end-to-end rate of log(m)/m. This would correspond to B=m(m-1)/2 pairs, and thus we get the same rate of 1/sqrt(B) of Theorem 5 (without even the intermediary of the larger sample size n). Additionally, Table 1 (in both the main text and supplements) and Figure 1 (in the supplements) do not appear to show a difference in rate. Thus it would seem that the main "damage" is to the constants, not to the rate as claimed. The only other claim in this regard (Lines 244-246 in the supplementary material) is that the gap widens with increased model complexity. This wider gap could simply be due to the fact that both excess risks get multiplied by the VC dimension (so it could be a wider additive gap, while the multiplicative gap in excess risk remains constant). Minor remarks: - In Line 92, I think what is meant is "Marginalized over the features" and not "Conditioned upon the features". Once the features are specified, the edges are independently generated, though not identically. This should be stated plainly. On the other hand, of course the data point triplets Z_{ij}=(X_i,X_j,e_{ij}) are not independent (though marginally identically distributed). - The term on Line 169, Lambda(g)=R(g)-R^*, may understandably be called excess risk. However, that terminology is often reserved for the excess over the best-in-model, rather than the Bayes risk. Especially given that the latter terminology is also used in the paper (Line 122), perhaps a different choice of words should be used on Line 169.

Confidence in this Review

2-Confident (read it all; understood it all reasonably well)


Reviewer 3

Summary

Predicting connections between a set of data points - named the graph reconstruction problem through empirical risk minimization is studied in the paper. This paper provides theoretical insights regarding to online learning. They proved a faster rate, using techniques such as VC dimension.

Qualitative Assessment

The paper contains solid results and is well-written.

Confidence in this Review

1-Less confident (might not have understood significant parts)


Reviewer 4

Summary

The paper focuses on a graph reconstruction problem. More specifically, the paper proves that a tight bound is always achieved by empirical reconstruction risk minimizers, and provides a sampling-based approach to approximate the reconstruction risk. This sampling-based approach introduces the incomplete graph reconstruction risk, which has much fewer terms. The paper also illustrates the theoretical results by numerical experiments on synthetic and real graphs.

Qualitative Assessment

This paper focuses on a graph reconstruction problem, proving a tight bound of ERM and proposing a fast graph scaling-up method. The scaling-up method provides a fast algorithm for the reconstruction problem and reveals some insights into the differences between pairs and vertices sampling. However, there are some problems: 1. The assumption that all vertices in graph are independent is somewhat impractical. Two vertices linked by a edge should be dependent in the real world. 2. line 198: the definition of degenerate U-statistic should not be a conditional expectation. 3. The "dataset spliting" strategy is proposed abruptly in section 3. How does this strategy relate to the tighter bound? 4. Some experiments should be done using different classifiers to reconstruct the graph to show the impact of model complexities on the ERM bound especially on the real network. 5. More related works should be discussed, e.g., learning from noniid data.

Confidence in this Review

3-Expert (read the paper in detail, know the area, quite certain of my opinion)


Reviewer 5

Summary

The paper adjusts fast rate analysis empirical risk minimization of U-statistics to graph reconstruction problem that is formularized in previous paper. And it proves that learning rate decreases from O( 1 / sqrt(n) ) to O( log(n) / n ) and any reconstruction rule “g” can be found in the improved learning time. In addition, by using incomplete U-statistics with sampling, the proposed method can work on the graph of large size which the previous work cannot work on.

Qualitative Assessment

1) Formularizing graph reconstruction problem and fast rate analysis part is similar with previous works. One of main contributions of the paper is to combine the two parts, however, proof is not enough to understand the contents and creativity. The proof goes along very quickly roughly, so it is difficult to understand it. It may increase the completeness with more detail proof in the supplementary material. 2) The learning process (graph reconstruction) in the paper is to find the “g” that minimizes the empirical risk “R”, or the excess reconstruction risk. But there is no training (learning) process for getting minimizer “g”. And the paper will be better, if showing the way how to utilize a learned “g”, not just graph reconstruction. 3) There is no comparison with previous research about graph reconstruction. I cannot estimate the performance about the proposed method, such as how low graph error reconstruction error is, how fast learning time is. Even though the previous work cannot work on the graph of large size which the proposed algorithm can work on, it is necessary to evaluate the performance by comparison using the graph relative small size which is maximum size for the previous work. By this comparison, the proposed method may be able to be more meaningful.

Confidence in this Review

2-Confident (read it all; understood it all reasonably well)


Reviewer 6

Summary

The authors provide fast learning rates of O(log n/n) for the problem of graph reconstruction. This improves upon the learning rates of O(1/\sqrt{n}) obtained by Biau and Bleakley (2006) who also developed the empirical risk minimization framework for graph reconstruction studied in this paper. Another contribution of the paper is showing that the approximate empirical risk computed by sampling (with replacement) a limited number of vertex pairs B, instead of computing the exact risk over all O(n^2) vertex pairs, has essentially the same rate bounds as the exact empirical risk minimizer --- with the parameter B controlling the tradeoff between convergence rate and computational complexity.

Qualitative Assessment

1) While the improved convergence rates for the problem of graph reconstruction derived in the paper does advance the literature of graph reconstruction, the result doesn't seem to be surprising or novel altogether. Such fast convergence rate of O(log n / n) have been obtained by Clemencon et at (2008) for the ranking problem. Indeed, the proofs in the paper follow the general proof structure of the Clemencon paper. The main trick seems to be showing that the low noise condition necessary for the ranking problem, in facts holds without any assumption (universally) for the problem of graph reconstruction. 2) The analysis of sampling based approximate empirical risk is a useful contribution of the paper in scaling up the inference to large graphs.

Confidence in this Review

1-Less confident (might not have understood significant parts)